

# Error sources in the retrieval of aerosol information over bright surfaces from satellite measurements in the oxygen A-band

Swadhin Nanda[1,2], Martin de Graaf[1], Maarten Sneep[1], Johan F. de Haan[1], Piet Stammes[1], Abram F. J. Sanders[3], Olaf Tuinder[1], J. Pepijn Veefkind[1,2], and Pieternel F. Levelt[1,2]

[1]Royal Netherlands Meteorological Institute (KNMI), Utrechtseweg 297, 3731 GA De Bilt, The Netherlands
[2]Delft university of Technology (TU Delft), Mekelweg 2, 2628 CD Delft, The Netherlands
[3]University of Bremen, Institute of Environmental Physics, Otto-Hahn-Allee 1, 28359 Bremen, Germany

*Correspondence to:* Swadhin Nanda (nanda@knmi.nl)

**Abstract.** Retrieving aerosol optical thickness and aerosol layer height over a bright surface from measured top of atmosphere reflectance spectrum in the oxygen A-band is known to be challenging, often resulting in large errors. In certain atmospheric conditions and viewing geometries, a loss of sensitivity to aerosol optical thickness has been reported in literature. This loss of sensitivity has been attributed to a phenomenon known as critical surface albedo regime, which is a range of surface albe-

dos for which the top of atmosphere reflectance has minimal sensitivity to aerosol optical thickness. This paper extends the concept of critical surface albedo for aerosol layer height retrievals in the oxygen A-band, and discusses its implications. The underlying physics are introduced by analysing top of atmosphere reflectance spectra obtained using a radiative transfer model. Furthermore, error analysis of the aerosol layer height retrieval algorithm are conducted over dark and bright surfaces to show the dependency on surface reflectance. The analysis shows that the information on aerosol layer height from atmospheric path

contribution and the surface contribution to the top of atmosphere are opposite in sign — an increase in surface brightness results in a decrease in information content. In the case of aerosol optical thickness, these contributions are anti-correlated, leading to large retrieval errors in high surface albedo regimes. The consequence of this anti-correlation is demonstrated with measured spectra in the oxygen A-band from GOME-2A instrument on board the Metop-A satellite over the 2010 Russian wildfires incident.

# 1 Introduction

Aerosols are one of the largest source of uncertainties in our understanding of the Earth's current climate and its future projection, because of the role they play in complex atmospheric processes that influence the Earth's radiation budget (IPCC, 2014). More generally, aerosols influence the climate either directly through absorption and scattering of solar radiation, or indirectly through cloud formation and aerosol-cloud interaction.

In climate studies, the direct radiative effect of aerosols is calculated to understand its net contribution to the Earth's total radiation budget. This depends on aerosol macrophysics (such as vertical distribution) and microphysics (such as size distribution and single scattering albedo), which determine if an aerosols in a particular scenario are more efficient in absorbing or scattering the incoming solar radiation and the thermal radiation from within the Earth's atmosphere. The ability of aerosols to





absorb radiation can influence thermal stability of the atmosphere, which in turn influences cloud formation and atmospheric chemistry (IPCC, 2014; Chung and Zhang, 2004). Knowledge on the vertical distribution on aerosols is, hence, an important piece of the puzzle to reduce uncertainties in our understanding of Earth's climate. Because of the high degree of variability of aerosols in both time and space, this knowledge is required at a high spatio-temporal resolution.

5    To observe (among other atmospheric parameters) aerosols, many space borne Earth observation initiatives have been proposed to monitor the Earth's atmosphere with either active or passive remote sensing techniques. An example of such an initiative is the Cloud-Aerosol LIdar with Orthogonal Polarisation (CALIOP) instrument on board NASA's Cloud-Aerosol Lidar and Infrared Pathfinder Satellite Observations (CALIPSO) mission, which provides information on the vertical distribution of aerosols. However, because of the limited swath of a space-borne lidar instrument, the mission coverage area is significantly reduced. This gap in the data can be filled with satellite missions carrying passive remote sensing instruments, which have a larger coverage area with good temporal resolution. One such initiative is the Copernicus programme by the European Commission (EC) partnered with ESA, which aims to provide accurate information of atmospheric composition from space. Of its missions, the Sentinel-5 Precursor, Sentinel-5 and Sentinel-4 are examples of polar orbiting and geostationary satellites equipped with hyperspectral sensors (Veefkind et al., 2012; Ingmann et al., 2012).

15    Hyperspectral instruments on board the Sentinel-4/5/5P missions measure Earth radiance and Solar irradiance in the top of atmosphere, spectrally resolved over a wide wavelength range. Of the wavelength bands measured, these instruments also measure in the oxygen A-band between 758 nm and 770 nm where absorption of solar radiation is dominated by molecular oxygen and its isotopologues. The presence of aerosols in the atmosphere significantly impacts absorption of solar radiation by molecular oxygen (Figure 1, left). In the absence of clouds and aerosols, the oxygen A-band can either be almost transparent or opaque to solar radiation, owing to the large variation in the absorption cross section within the spectra. In the presence of an aerosol layer in the atmosphere, the absorption intensity of the spectra can provide useful vertical information (as observed in Figure 1, right) — deeper absorption lines correspond to a lower aerosol layer, shallow absorption lines for a higher aerosol layer. This is the basis of retrieving aerosol layer height from the oxygen A-band. Currently, the Copernicus Sentinel-4/5/5-P aerosol layer height algorithms are designed to exploit oxygen absorption spectra in the A-band to retrieve the height of an aerosol layer.

The retrieval of aerosol properties from the oxygen A-band presents a few challenges, one of them being that aerosol layers in the atmosphere are usually optically thin, and are quite difficult to observe in the presence of clouds. This is because clouds have an optical depth which is typically orders of magnitude larger than that of aerosols, and are more efficient in scattering incoming radiation. Consequently, aerosol retrieval algorithms generally refrain from retrieving over cloudy scenes; our algorithm is no exception to this and requires cloud screening to filter out pixels containing clouds.

While cloudy pixels can be filtered out to a certain degree, retrieving aerosols from measurements in the oxygen A-band over bright surfaces faces a host of other challenges. From literature, it is understood that aerosol information content from measured spectra in the oxygen A-band reduces as the surface albedo increases (Corradini and Cervino, 2006; Sanghavi et al., 2012). Sanders et al. (2015) report potentially large biases in their aerosol layer height retrievals from the oxygen A-band when the surface albedo is fitted. In a previous paper, Sanders and de Haan (2013) also report that certain specific combinations of





geometry, aerosol, and surface properties can result in unusually large uncertainties in the retrieved aerosol layer height (see also Figure 8-2 in Sanders and de Haan (2016)). Such large biases can perhaps be attributed to a phenomenon known as the critical surface albedo regime (Seidel and Popp, 2012), wherein for specific surface albedos, the top of atmosphere reflectance becomes independent of the aerosol optical thickness. Sanders et al. (2015) observe that when the surface albedo isn't fitted, typical uncertainties in the surface albedo database over land can result in large biases. From our analyses, we understand that for relative errors up to 10% in the surface albedo, retrievals over dark surfaces are non-consequential, whereas the same over sufficiently bright surfaces (surface albedo greater than 0.2) can suffer from very large biases.

A combination of all the error sources discussed previously can result in large biases. In fact, we observe that the presence of such biases often lead to no convergence, with no concrete predictability on which pixel is likely to yield no result. Because of this, the operational algorithm wastes resources trying to retrieve aerosol layer height from pixels that potentially do not have any usable aerosol information. This is especially problematic in the framework of high resolution instruments, which demand operational processors to make efficient use of computational time and effort to process large number of spectra (typically several hundred per second). In order to design more efficient operational algorithms, the cause of these errors needs to be extended beyond the framework provided by Seidel and Popp (2012) into the oxygen A-band for aerosol optical thickness as well as aerosol layer height.

This paper analyses simulated top of atmosphere reflectance spectra in the oxygen A-band and provides an explanation for the loss of aerosol information over bright surfaces. Its implication is provided in an optimal estimation framework, specific to the retrieval of aerosol layer height, with results from sensitivity analyses. The analysis is followed up with a demonstration in a real data environment by retrieving aerosol layer height over a bright surface. The case study chosen is the retrieval of optically thick biomass burning aerosol plumes over the 2010 Russian wildfires, to demonstrate the effect of this loss of aerosol information over land. This paper is one in a series of papers on development of an operational oxygen A-band Aerosol Layer Height retrieval algorithm for Sentinel-4/5/5-P by KNMI, preceeded by Sanders and de Haan (2013) and Sanders et al. (2015). The current operational ALH algorithm for S5P is described in Sanders and de Haan (2016). While the results of this paper are relevant for the Sentinel 5-Precursor algorithm as well, the instrument model used in the sensitivity studies is for the UVN spectrometer on the S4 mission.

The next section (Section 2) provides a description of the forward model and the optimal estimation framework. Section 3 discusses the concept of aerosol-surface ambiguities in the oxygen A-band. Section 4 describes various sensitivities of our retrieval algorithm focusing on the difference between dark and bright surfaces. Section 5 discusses aerosol layer height retrievals over the 2010 Russian wildfires using GOME-2A data. Section 6 concludes this paper with a discussion and the implication of the findings from this paper.





## 2 The forward model and the inverse method

### 2.1 Forward model

There are three primary parts of the forward model, namely the the atmospheric model, the radiative transfer code, and the instrument model. A radiative transfer code is used to model a high resolution top of atmosphere radiance by propagating

radiation through the atmosphere described by the atmospheric model. The top of atmosphere reflectance $R$ computed by the forward model is defined as the ratio of the radiance $I$ of the pixel measured by the instrument to the top of atmosphere solar irradiance $E_0$ of the pixel on a horizontal surface unit,

$$R(\lambda) = \frac{\pi I(\lambda)}{\mu_0 E_0(\lambda)}. \tag{1}$$

$\mu_0$ represents the cosine of the solar zenith angle of the pixel, and $\lambda$ represents the wavelength.

The top of atmosphere reflectance is calculated after the measured radiance and irradiance are convolved with the Instrument Spectral Response Function (ISRF) of the hyperspectral sensor in order to simulate measured spectra by a satellite instrument. For simulations, the high resolution solar spectra by Chance and Kurucz (2010) is used.

### 2.1.1 Radiative transfer model

The radiative transfer model is the Layer Based Orders of Scattering (LABOS) method, which is a variant derived from the

Doubling-Adding method (de Haan et al., 1987). Atmospheric properties are calculated line-by-line to compute the reflectance at the top of atmosphere. The radiative transfer code is a part of a software package called DISAMAR (Determining Instrument Specifications and Analysing Methods for Atmospheric Retrievals), which is the main workhorse of operational algorithm development efforts at KNMI for oxygen A-band aerosol height retrieval with S5P/S4/S5 instruments. Scattering by gases is described by Rayleigh scattering, which has a low scattering cross section in this wavelength region. Because of this, polarisa-

tion is ignored. Wavelength shifts caused by rotational Raman scattering (RRS) are ignored in order to reduce computational effort, since line by line calculations are computationally expensive in the oxygen A-band. This is convenient, since the Raman scattering cross section is even smaller than that of Rayleigh scattering. The atmosphere in the forward model is plane-parallel for the Earth radiance, and spherically corrected for the incoming solar irradiance.

### 2.1.2 Atmospheric model

For cloud-free conditions, the following four absorption and scattering processes are significant in the wavelength range between 758 nm and 770 nm: scattering by gases, reflection of light by the surface, scattering and absorption by aerosol particles, and absorption by molecular oxygen. Absorption of solar radiation by $O_3$ and $H_2O$ are ignored, since they are not dominant absorbing gases in this spectral range.

    The surface reflectance is assumed isotropic, described by its albedo. Depending on the surface albedo, a surface can either

be bright or dark. Dark surfaces are classified with surface albedo close to 0.05 (or lower), which in the oxygen A-band spectral



region typically corresponds to ocean surfaces. Bright surfaces in the oxygen A-band on the other hand have a surface albedo of 0.2 (intermediately bright) and higher and are primarily over land. For the oxygen A-band at 760 nm, typical values of surface albedo over vegetated surfaces exceed 0.4 due the wavelength band lying beyond the red edge where absorption of solar radiation by chlorophyll diminishes. Scenes with snow or ice are not processed.

Aerosols are represented as a single layer with a fixed pressure thickness of 50 hPa, containing aerosol particles with a fixed aerosol optical thickness and aerosol single scattering albedo. Aerosol layer height is defined as the mid-pressure of the aerosol layer — if the aerosol layer extends from 650 hPa to 600 hPa, the aerosol layer height is 625 hPa. In the operational S5P aerosol layer height algorithm, currently the aerosol phase function is a Henyey-Greenstein model (Henyey and Greenstein, 1941) with an asymmetry factor of 0.7, and an aerosol single scattering albedo of 0.95 (Sanders et al., 2015). While a Mie

scattering model could be used instead of the Henyey-Greenstein, the latter is computationally less expensive and hence more optimal for the operational algorithm.

Oxygen absorption cross-sections are derived from the NASA JPL database, following Tran and Hartmann (2008) who indicate that line parameters in the JPL database are more accurate than the HITRAN 2008 database. First-order line mixing and collision induced absorption by $O_2$-$O_2$ and $O_2$-$N_2$ are derived from Tran et al. (2006) and Tran and Hartmann (2008).

### 2.1.3  Instrument model

The instrument model is described by the instrument slit function, whose spectral resolution depends on its Full Width at Half Maximum (FWHM), and its noise model. For this study, oxygen A-band is simulated using specifications of the Sentinel-4 Ultraviolet Visible and Near infrared (UVN) instrument, which is set to launch in 2022. The instrument's platform has been designed as a geostationary atmospheric sounder with a hourly coverage over Europe and Northern Africa at a spatial

resolution of $8 \times 8$ km$^2$ sampled at 45°N and 0°E. The near infrared spectrometer has a FWHM of approximately 0.116 nm in the near infrared, oversampled by a factor of 3. Effectively, the spectral sampling interval of the instrument is 0.04 nm. Aerosol layer height will be an operational product provided by the Sentinel-4 mission. An example of oxygen A-band spectra at a 0.116 nm resolution is provided in Figure 1. For retrievals with real data, measurements from the Global Ozone Monitoring Experiment-2 on board the MetOp-A satellite are used. Launched on October 16, 2006, GOME-2A is an optical spectrometer

fed by a scanning mirror which enables across-track scanning in the nadir. The instrument has a spectral sampling interval of approximately 0.21 nm at 758 nm (spectral resolution of 0.48 nm for channel 4), and has a nominal spatial resolution of $80 \times 40$ km$^2$ (Munro et al., 2016). The noise model assumes a noise spectrum dominated by shot noise.

### 2.2  Inverse method

The inverse method is based on the Optimal Estimation (OE) framework described by Rodgers (2000), which is a Maximum A-

Posteriori (MAP) estimator that constrains the least-squares solution with a-priori knowledge on the state vector. The method





assumes Gaussian statistics for the a-priori errors. The iterative method is a Gauss-Newton approach, and the estimation parameters are the aerosol optical thickness $\tau$ and the aerosol layer height $z$. The cost function $\chi^2$ is defined as,

$$\chi^2 = [\mathbf{y} - \mathbf{F}(\mathbf{x}, \mathbf{b})]^T \mathbf{S}_\epsilon^{-1} [\mathbf{y} - \mathbf{F}(\mathbf{x}, \mathbf{b})] + (\mathbf{x} - \mathbf{x_a})^T \mathbf{S_a}^{-1} (\mathbf{x} - \mathbf{x_a}),$$ (2)

where $\mathbf{y}$ is the measured reflectance, $\mathbf{F}(\mathbf{x}, \mathbf{b})$ is the vector of calculated reflectance using the forward model, $\mathbf{x}$ is the state vector
containing fit parameters, $\mathbf{b}$ is the vector containing other model parameters, $\mathbf{S}_\epsilon$ is the measurement error-covariance matrix, $\mathbf{x_a}$ is the a-priori state vector, and $\mathbf{S_a}$ is the a-priori error-covariance matrix. The matrices $\mathbf{S_a}$ and $\mathbf{S}_\epsilon$ are diagonal, and are not correlated. $[\mathbf{y} - \mathbf{F}(\mathbf{x}, \mathbf{b})]^T \mathbf{S}_\epsilon^{-1} [\mathbf{y} - \mathbf{F}(\mathbf{x}, \mathbf{b})]$ is the measurement part of the cost function, whereas $(\mathbf{x} - \mathbf{x_a})^T \mathbf{S_a}^{-1} (\mathbf{x} - \mathbf{x_a})$ is the a-priori part of the cost function.

The a-posteriori error covariance matrix $\hat{\mathbf{S}}$ is computed as,

$$\hat{\mathbf{S}} = (\mathbf{K}^T \mathbf{S}_\epsilon \mathbf{K} + \mathbf{S_a}^{-1})^{-1},$$ (3)

where $\mathbf{K}$ is the Jacobian with its columns containing partial derivatives of the reflectance with respect to the state vector elements. DISAMAR calculates the Jacobian semi-analytically, similar to the reciprocity method described by Landgraf et al. (2001). The Jacobian drives the retrieval towards the solution as an integral component in the update to the state vector,

$$\mathbf{x}_{n+1} = \mathbf{x_a} + (\mathbf{K_n}^T \mathbf{S}_\epsilon^{-1} \mathbf{K_n} + \mathbf{S_a}^{-1})^{-1} \mathbf{K_n}^T \mathbf{S}_\epsilon^{-1} [\mathbf{y} - \mathbf{F}(\mathbf{x}_n) + \mathbf{K_n}(\mathbf{x_n} - \mathbf{x_a})],$$ (4)

where $\mathbf{x}_{n+1}$ is the next iteration to the $n^{th}$ iteration in the retrieval, and $\mathbf{K_n}$ is the Jacobian evaluated at the $n^{th}$ iteration. The Jacobian is also the primary reason why the retrieval can fail — the Jacobian can become singular if the value of the partial derivative of the reflectance to the a state vector parameter is very low, or is correlated to another parameter in the state vector. In these cases, the error covariance matrix does not exist, since the inverse covariance matrix is non-invertible; if it is *nearly* singular, the problem is ill-conditioned and may result in very large biases in the estimation.

The inverse method reaches a solution if the change in the state vector between iterations is below a convergence threshold. It is possible that during iterations, the inverse method estimates state vector elements beyond their physical boundary conditions. In such a case, the state vector element is adjusted back to just within its physical limits. If the adjustment is made in two consecutive iterations, the retrieval is stopped and no solution is reached. The upper cap in the number of iterations is set at 12, beyond which the retrieval is said to have failed. In this paper, these failed retrievals are termed as non-convergences. The next
section discusses the atmospheric conditions that can potentially lead to these non-convergences.





## 3 Aerosol-surface ambiguities in the oxygen A-band

### 3.1 Influence of surface reflectance on aerosol information content in the oxygen A-band

The top of atmosphere reflectance over a surface with an albedo $A_s$ can be written as the sum of photon path contribution $R_p$ and surface contribution $R_s$,

$$R(\lambda, A_s) = R_p(\lambda) + R_s(\lambda, A_s). \tag{5}$$

$R_p$ is the top of atmosphere reflectance in the absence of a surface. $R_s$ is calculated by subtracting the path contribution from the total top of atmosphere reflectance, and represents contributions from photons that have been reflected one or more times by the surface. $R_s$ is dependent on the absorbing and scattering species present in the atmosphere, and also includes aerosol influences. $R_p$ is calculated by substituting $A_s = 0.0$ and calculating the top of atmosphere reflectance in DISAMAR. $R_s$ is calculated by subtracting $R_p$ from $R$. With increasing viewing angle, $R_p$ increases whereas $R_s$ decreases (Figure 2). This is in line with expectation, since the slant aerosol optical thickness increases, which increases the amount of contribution that aerosols have in $R(\lambda, A_s)$. At steeper geometries, light at the top of atmosphere is more diffuse than direct, which is the primary reason why $R_s$ decreases.

For a model parameter $x$ with two values $x_a$ and $x_b$, the difference spectra $\Delta R_{\Delta x}$ , defined as

$$\Delta R_{\Delta x} = R_{x_a} - R_{x_b}, \tag{6}$$

can reveal the influence the model parameter $x$ has on the oxygen A-band. The spectral shape of $\Delta R_{\Delta x}$ can also show parts of the spectrum that are more sensitive to $x$. Following Equations 5 and 6, $\Delta R_{\Delta x}(\lambda, A_s)$ is defined as

$$\Delta R_{\Delta x}(\lambda, A_s) = \Delta R_{p_{\Delta x}}(\lambda) + \Delta R_{s_{\Delta x}}(\lambda, A_s). \tag{7}$$

If $\Delta R_{p_{\Delta x}}$ and $\Delta R_{s_{\Delta x}}$ have opposing signs, it reveals an interference between these two contributions to the top of atmosphere reflectance, which may result in a reduction of sensitivity to the parameter $x$. In such a case, the relative difference of the magnitude between $\Delta R_{p_{\Delta x}}$ and $\Delta R_{s_{\Delta x}}$ gives an idea on the magnitude of interference.

Comparing $\Delta R_{p_{\Delta z}}$ and $\Delta R_{s_{\Delta z}}$ at two different aerosol layer heights ($z$) for two different scenes with the same atmospheric conditions (Figure 3, left panel), it is observed that $\Delta R_{p_{\Delta z}}$ and $\Delta R_{s_{\Delta z}}$ have opposite signs and $R_p$ is relatively more sensitive to aerosol layer height than $R_s$. This is especially the case in the deepest part of the R-branch between 759.50 nm and 761.30 nm and parts of the P-branch between 761.30 nm and 763.00 nm, where the higher absorption cross section reduces the number of photons that can reach the surface. This ultimately reduces the magnitude of $R_s$ to the top of atmosphere for these absorption sub-bands. $\Delta R_{s_{\Delta z}}$ over ocean and vegetation also shows an increase in its overall magnitude with an increase in surface albedo, and hence an increase in interference between $\Delta R_{p_{\Delta z}}$ and $\Delta R_{s_{\Delta z}}$. Figure 4 represents the variation of the




derivative of reflectance with respect to aerosol properties, for increasing surface albedo. Albeit sublte, the consequence of interference between $\Delta R_{p_{\Delta z}}$ and $\Delta R_{s_{\Delta z}}$ is observed in Figure 4 (Top), where $\partial R/\partial z$ for the deepest part in the R-branch and parts of the P-branch diminishes gradually with an increase in surface albedo.

The same experiment is repeated for aerosol optical thickness ($\tau$), and the results are presented in Figure 3 (middle panel).

$\Delta R_{p_{\Delta\tau}}$ and $\Delta R_{s_{\Delta\tau}}$ are anti-correlated (Pearson correlation coefficient is -0.99, irrespective of the surface albedo), and the magnitude of $\Delta R_{s_{\Delta\tau}}$ increases with an increase in surface albedo. Figure 4 (Middle) shows the partial derivative of the reflectance with respect to $\tau$ for increasing surface albedo. The interference between $\Delta R_{p_{\Delta\tau}}$ and $\Delta R_{s_{\Delta\tau}}$ explains negative derivatives in the higher surface albedo regime.

$\Delta R_{p_{\Delta\omega}}$ and $\Delta R_{s_{\Delta\omega}}$ of aerosol single scattering albedo ($\omega$) in Figure 3 (right panel) reveals a strong correlation (with a

Pearson correlation coefficient of almost unity). This suggests that an increase in surface albedo increases the sensitivity of the model to $\omega$. We suspect that this information predominantly arises from interactions between scattered light by aerosols and surface. The magnitude of the partial derivative of reflectance with respect to $\omega$ for increasing surface albedo (shown in Figure 4, bottom) shows an increase, which is in line with our analysis of Figure 3 (right panel).

For increasing surface albedo, the more dynamic parts of the $\partial R/\partial\tau$ spectrum in Figure 4 (Middle) correspond to spectral

points with less absorption by molecular oxygen. These are also the parts of the spectrum with a high signal to noise ratio (SNR) and high $S_\epsilon^{-1}$. From Equation 4, the inverse method gives a higher priority to spectral points with a higher $S_\epsilon^{-1}$. Intuitively, low information of $\tau$ from the oxygen A-band spectrum will increase the dependency of the inverse method to prior information. This is further discussed in the next section.

### 3.2   Aerosol-surface interplay in the top of atmosphere reflectance

In the inverse method, an a-priori error of 100% is assumed for the aerosol optical thickness, which gives it freedom to vary during iterations. If the first guess of the aerosol optical thickness is far from the solution, a large a-priori error ensures that the retrieval can estimate the parameter in fewer iterations. However, whether the Gauss-Newton optimisation reaches the correct solution depends on two primary factors, i. if the cost function has a global minimum, and ii. the the gradient of the cost function is sufficiently large, such that it is minimised significantly at every iteration.

From our analysis of $\Delta R_{\Delta x}$ for aerosol parameters, we have identified aerosol optical thickness to be the parameter most affected by an increasing surface albedo, due to the anti-correlation between $\Delta R_{p_{\Delta\tau}}$ and $\Delta R_{s_{\Delta\tau}}$. Because of this, the top-of-atmosphere reflectance spectrum becomes independent of aerosol optical thickness for higher surface albedo regimes (Figure 5).

Over a dark surface such as the ocean, top of atmosphere reflectance in the continuum is unique at different aerosol loads

(Figure 5, left panel). The variation in the top of atmosphere reflectance in the continuum reduces as the instrument points more towards the nadir. In such geometries, $R_s$ can play a more significant role than $R_p$ (Figure 2, blue line) and reduce the available information on $\tau$ in the $R(\lambda, A_s)$ spectrum. For bright surfaces, the variation in the the top of atmosphere reflectance spectrum is less for steeper geometries relative to the same geometries over the ocean (Figure 5, middle panel, green and blue line). There can also be cases where, provided sufficiently high aerosol loading, the top of atmosphere reflectance spectrum





in the continuum can be independent of aerosol optical thickness over very bright surfaces such as vegetation (Figure 5, right panel, green line). In such cases, more than one values of $\tau$ result in the same top of atmosphere reflectance. Henceforth in this paper, this phenomenon is termed as aerosol-surface ambiguity.

A loss in aerosol information can have special implications in the minimisation of the cost function. As observed in Figure
6, for lower surface albedo regimes there exists a single minima of the cost function. For such scenes, if the a-priori aerosol optical thickness is far from the true value, the gradient is sufficiently large such that a small change in the state vector between iterations leads to a significant minimisation of the cost function. As the surface albedo increases, this gradient decreases significantly, and can also result in the presence of multiple minima in the cost function (Figure 6, right). This makes the retrieval dependent on the initial guess of $\tau$. Also, as $R_s$ increases, the global minimum shifts away from the true $\tau$. This is
predominantly observed in Figure 6 (left, red line) over the bright surface for a viewing angle close to nadir, where $R_s$ is more dominant. For the same angle, the global minimum over a dark surface is situated at the true $\tau$ value. As the viewing angle increases over the bright surface, $R_p$ increases and the global minimum of the cost function moves closer towards the true $\tau$.

If the a-priori error assigned to aerosol optical thickness is large, presence of aerosol-surface ambiguities can result in non-convergences. Because the a-priori part of the cost function has a smaller value than the measurement part, reducing a-priori
error assigned to the aerosol optical thickness does not necessarily guarantee a solution to this issue since it does not remove the multiple-minima present in the cost function. Since errors between aerosol optical thickness and aerosol layer height are correlated (Sanders et al., 2015), a large error in the optical thickness will lead to a large error in the aerosol layer height estimate. The next section discusses the sensitivity of the aerosol layer height algorithm to this phenomenon by introducing model errors in a simulation environment.

**4   Error analysis**

In DISAMAR, forward models for simulation and retrieval have been kept separate so that errors can be introduced into the simulated spectra to mimic errors in a real retrieval scenario. In this section, the instrument model of the Sentinel-4 UVN near infrared spectrometer is used. The wavelength range for simulations and retrievals is between 758 nm and 770 nm. Error analysis is done for the aerosol layer height retrieval algorithm and a comparison is made between retrievals over ocean ($A_s =$
0.03) and land ($A_s = 0.25$, and $A_s = 0.4$). Bias in the aerosol layer height is defined as the difference between true and retrieved aerosol layer height (in hPa) — a negative sign indicates that the aerosol layer is retrieved closer to the ground. The aerosol layer height retrieved is a single layer for the entire atmospheric column, with a fixed thickness of 50 hPa.

**4.1   Sensitivity to model error in the aerosol layer thickness**

In a typical real-world scenario, aerosol plumes can be as thick as 200 hPa in the atmosphere, or more. We simulate a scene
containing an aerosol layer that extends approximately from the surface (1000 hPa) to 800 hPa in the atmosphere. The true $\tau$ is 1.0, and the a-priori $\tau$ is 0.5. The a-priori value of the aerosol layer height is 650 hPa, and the aerosol layer thickness is fixed at 50 hPa. In an ideal retrieval instance, the retrieved aerosol layer height (which has a thickness of 50 hPa) should coincide





with the height of the simulated thicker aerosol layer. We observe that, in general, the error in the retrieved aerosol layer height reduces as the viewing zenith angle increases (Figure 7, top left). This is explained by the lowered interference between $R_p$ and $R_s$ (Figure 2, red line). This is why the difference in errors between retrievals over the different surfaces reduces with an increase in viewing zenith angle.

At lower viewing zenith angles, the difference in aerosol layer height errors between retrievals over the different surfaces is the largest, since the effect of $R_s$ interfering with $R_p$ is significantly larger (Figure 2, blue line), which increases with an increase in surface albedo (Figure 3, left). The retrieved aerosol layer is biased towards the surface in all three surface albedo scenarios, with the aerosol layer being placed closer to the surface if the surface albedo is brighter. While it would appear that somehow the sensitivity of the retrieval to aerosol layer thickness increases with increasing surface albedo, this behaviour

is explained better as perhaps the most interesting consequence of the interference between $R_s$ and $R_p$ ingrained within the optimal estimation framework. To explain its mechanism, the following three inferences are highlighted:

- A look into the Jacobian in Figure 4 (top) shows that, for the same atmospheric conditions and the same aerosol layer height, an increase in surface albedo can reduce the magnitude of $\partial R/\partial z$ (parts of the P-branch between 762 nm and 765 nm, where absorption by oxygen is minimal). A reduced magnitude $\partial R/\partial z$ for a spectral point signifies a reduced

sensitivity to aerosol layer height at that spectral point.

- Figure 3 (left) shows low interference between $\Delta R_{p_{\Delta z}}$ and $\Delta R_{s_{\Delta z}}$ at spectral points with high absorption by oxygen, and vice versa. This suggests that, while parts of the spectrum with high absorption remain more-or-less the same irrespective of an increase in surface albedo, the parts with lower absorption get altered. From Figure 1 (right), it is observed that aerosol layers situated lower in the atmosphere have relatively deeper absorption lines at spectral sub-bands with low

absorption by oxygen.

- Assuming shot noise, parts of the spectrum with a lower absorption by oxygen have a higher SNR, and hence a higher $S_\epsilon^{-1}$ than parts of the spectrum with a higher absorption by oxygen. From Equation 4, a higher weight is given to these points in deciding the update to the state vector and in the overall optimal estimation framework.

    Because of the interference between $R_p$ and $R_s$, parts of the oxygen A-band spectrum with high SNR (and hence a higher

weight in the optimal estimation) appear to have more absorption for observations over high surface albedos, corresponding to an aerosol layer closer to the surface than the true aerosol layer height. Another consequence of retrieving aerosol layer height in the presence of interference between $R_s$ and $R_p$ is that the retrieval may become more susceptible to model error in aerosol and surface properties, such as the aerosol phase function anisotropy factor $g$, the aerosol single scattering albedo $\omega$ and especially the surface albedo $A_s$, which are fixed in the model.

**4.2   Sensitivity to model error in the aerosol phase function**

The presence of a model error in the aerosol phase function can result in large biases if the surface is bright (Figure 7, top right). For a higher surface brightness and a viewing angle close to nadir, this bias is larger. As the viewing angle increases,





the biases reduce significantly. The correlation of bias with surface albedo suggests that a biases cause by model errors are exacerbated by the surface contribution $R_s$, which reduces as viewing angle increases (Figure 2, right).

### 4.3  Sensitivity to model error in aerosol single scattering albedo

From Figure 4, aerosol single scattering albedo plays an increasingly significant role in the retrieval of aerosol layer height as
the surface gets brighter. Because of this, a mis-characterisation of aerosol single scattering albedo in the model can lead to very large biases over bright surfaces (Figure 7, bottom left), and also non-convergences. This is not the case for retrievals over the ocean, since the influence of aerosol single scattering albedo on the oxygen A-band spectrum is low. It is observed that, as the viewing angle increases, these biases drop significantly. This is again attributed to the decreased interference between $R_p$ and $R_s$ with increasing viewing angle.

### 4.4  Sensitivity to model error in surface albedo

Surface albedo is a critical component in the accurate retrieval of aerosol layer height over bright surfaces. Because it is a fixed parameter in the forward model, an error in the surface albedo can result in large biases in the retrieval. To simulate model errors, relative errors of -10% to 10% are introduced in the retrieval forward model, such that the surface is modeled darker or brighter than the true value. For relative errors of $\pm10\%$, the retrieved aerosol layer height can be biased more than two
orders of magnitude larger over land than the over the ocean (Figure 7, bottom right). For retrievals over a bright surface such as vegetation ($A_s = 0.4$ or greater), the model error can result in non-convergences. As the model error reduces, retrievals over land with a surface albedo of 0.25 become more acceptable. However, over very bright surfaces, an inaccuracy in surface albedo of more than 2% can result in biases greater than 100 hPa.

    The sign of aerosol layer height retrieval biases is dependent on the sign of the error in the surface albedo. If the a-priori
surface albedo is greater than the true surface albedo, the aerosol layer height is estimated much closer to the ground. From our analysis of splitting the oxygen A-band spectrum into $R_p$ and $R_s$, we understand that this has to do with the increased $R_s$ due to the surface albedo fixed at a higher value. The next section demonstrates the implication of these errors in a real retrieval scenario.

### 5  Demonstration case: 2010 Russian wildfires

The 2010 Russian wildfires began in late July and lasted for several weeks until the beginning of September. Literature reports droughts and record summer temperatures in the same year as a precursor to the wildfires, both of which have been attributed to climate change (Hansen et al., 2012). A consequence of the forest fires were optically thick aerosol plumes over the country, especially over Moscow. In the first few weeks of August, 2010, due to the presence of a strong anti-cyclonic circulation pattern in the atmosphere, the impact of biomass burning aerosols on air quality in Moscow was markedly larger than what
was observed from previous wildfire incidences — the UV Aerosol Index (AI) reported by the Ozone Monitoring Instrument



(OMI) on board the NASA Aura mission observed an increase by a factor of 4.1 from previous years (Witte et al., 2011) over Moscow, due to aerosol plumes originating from the South and East of the city.

The aerosol plume above Russia on the 8th of August, 2010 serves as a test case for the aerosol layer height retrieval algorithm, due to fairly cloud-free conditions and the optical thickness of the aerosol plume (see Figure 8, right). Because of this, we do not employ a cloud-screening method. The GOME-2A instrument crosses over the scene at approximately 09:45 hrs - 09:47 hrs at local time. The GOME-2A pixels within the region of interest are recorded between 0745 hrs UTC and 0748 hrs UTC, at approximate latitude bounds of 52° and 60° and longitude bounds 29° and 45°. This corresponds to 255 pixels in total. Meteorological information relevant to the retrieval are temperature-pressure profiles and surface pressure, acquired from the European Center for Medium-Range Weather Forecast (ECMWF) ERA-Interim database (Dee et al., 2011) at the GOME-2A pixel using nearest neighbour interpolation. Surface albedo is derived using nearest neighbour interpolation from Tilstra et al. (2017), who provide monthly Lambertian Equivalent Reflectivity (LER) climatologies on a 0.5° x 0.5° grid. Typical values of the surface albedo over the region of interest is around 0.21. In the inverse method, the first guess of the aerosol layer height is approximately 800 hPa. The a-priori aerosol optical thickness is 1.0 at 760 nm.

CALIOP data is used for validation, which provides vertical distribution of aerosols and clouds for a footprint of approximately 70 m, with a 5 km horizontal resolution (Winker et al., 2009). While the coverage of the instrument is not as expansive as the GOME-2 instrument, the level of information available from CALIOP gives a good idea on the vertical position of aerosols in the atmosphere. For a better validation dataset, CALIOP data recorded between coordinates 52.0° latitude and 64.0° latitude, approximately around 1045 hrs UTC is used for comparison of GOME-2A aerosol layer height retrieval results. The Level-1 CALIOP attenuated backscatter data from 1064 nm is used because lidar in the visible region (532 nm) can get heavily attenuated over optically thick plumes. As can be seen from Figure 9, the aerosol layer is situated in between the surface and 5 km above the surface. In total, 82 GOME-2A pixels falling within 100 km of the CALIPSO track are considered for comparison.

The operational algorithm retrieves aerosol layer height and aerosol optical thickness, with fixed a-priori values, as mentioned in Table 1. Following evaluation of the algorithm on GOME-2A pixels by Sanders et al. (2015), the surface albedo is not included in the state vector. The single scattering albedo is not fitted in the sensitivity analyses in order to maintain consistency with the current operational algorithms for the Sentinel missions, which currently do not fit this parameter.

## 5.1 Results from the retrieval algorithm

Out of the chosen 255 GOME-2A pixels, 155 pixels converged and 100 pixels failed to converge to a solution (40% of the pixels do not converge). The algorithm retrieved aerosol layers primarily in the lower troposphere, roughly within 0 - 3 kilometers (Figure 8, left). The mean aerosol layer height retrieved is 714 m above the ground with a standard deviation of 647 m and a median of 450 m. The retrieved aerosol layers are optically thick (Figure 8, middle), with an mean retrieved aerosol optical thickness of 3.0, a standard deviation of 1.8, and a median of 2.5. The retrievals over the primary aerosol plume do not converge to a solution.





**Table 1.** A-priori and validation information required to process data over 2010 Russian wildfires on the $8^{th}$ of August, 2010.

| parameter | source | remarks |
|---|---|---|
| radiance and irradiance | GOME-2A | data between latitudes 52° and 60° and longitudes 29° and 45° (255 pixels) |
| solar and satellite geometry | GOME-2A Level 1-b data | |
| surface albedo $A_s$ | Tilstra et al. (2017) | GOME-2 LER at 0.5° x 0.5° grid at 758 nm and 772 nm |
| surface pressure $p_s$ | ERA-Interim | |
| temperature pressure profile | ERA-Interim | |
| aerosol optical thickness $\tau$ | | state vector element, a-priori = 1.0 |
| aerosol layer height $h_{mid}$ [km] | | state vector element, a-priori = $p_s$ - 200 hPa |
| aerosol single scattering albedo $\omega$ | | fixed at 0.95 |
| aerosol phase function $P(\theta)$ | | Henyey-Greenstein with asymmetry factor $g$ of 0.7 |
| cloud mask | | none |
| validation | CALIOP lidar profiles | 5 km × 5 km total attenuated backscatter at 1064 nm |

Figure 9 (top) provides results of retrieving aerosol layer height over the chosen 82 GOME-2A pixels colocated to the CALIPSO track. The CALIOP backscatter data shows that the aerosol plume extends from the ground to approximately 4 km between latitudes 53° and 60°. Beyond 60° latitude, the aerosol layer is elevated. Of the 82 pixels, 52 converge to a solution. From Figure 9, it is observed that the retrieved aerosol layer heights are generally biased closer to the surface. From our analysis in Figure 7 (top left), we understand this to be a consequence of the interference between $R_p$ and $R_s$ in the presence of model error in the aerosol layer thickness.

In Figure 9, the retrieval does not converge to a solution between latitudes 57° and 60°. This area also corresponds to the primary biomass burning plume in Figure 8. However, the estimated aerosol layer height in the last iteration for these pixels seems to be located within the aerosol plume (Figure 9, top, white crosses between latitudes 57° and 60°). To investigate this, we retrieve $\tau$ from the top-of-atmosphere reflectance in the continuum with different a-priori optical thickness values in order to test whether the non-uniqueness of aerosol optical thickness is a potential cause of retrieval non-convergence.

## 5.2 Retrieving aerosol layer height with multiple a-priori aerosol optical thickness values

Aerosol optical thickness ($\tau$) is first retrieved from the continuum before the oxygen A-band between 755 nm - 756 nm. $\tau$ is retrieved with two a-priori values $\tau_a$ and $\tau_b$. In these retrievals, the aerosol layer height is kept fixed at any arbitrary value, since its value will hardly affect the continuum.



First, $\tau_a = 1.0$ is chosen, and the retrieved solution $\tau_a^{'}$ is then used to decide the a-priori value $\tau_b$. If the solution for $\tau_a^{'}$ is not reached, then $\tau_b^{'}$ is not calculated. In the case that $\tau_a^{'}$ is retrieved, $\tau_b$ is chosen in the following manner,

$$\tau_b = \begin{cases} \tau_a^{'}/2 & \text{if } \tau_a^{'} < \tau_a \\ \tau_a^{'} + 0.5 & \text{if } \tau_a \leq \tau_a^{'} < 10.0. \end{cases} \tag{8}$$

If the retrieval for $\tau_b^{'}$ fails, then we can infer a dependence on a-priori information. If the retrieval is successful, $\tau_a^{'}$ and $\tau_b^{'}$

are compared to check if they are similar using the following criterion,

$$\tau_a^{'} \approx \tau_b^{'} \text{ if } \mathbf{abs}(\tau_a^{'} - \tau_b^{'}) < T \times \mathbf{min}(\tau_a^{'}, \tau_b^{'}), \tag{9}$$

where $T$ is a threshold, chosen to be 0.15. Increasing this threshold increases the margin of similarity of $\tau_a^{'}$ and $\tau_b^{'}$. This method is henceforth called the prefit method.

Applying the prefit method to the GOME-2A pixels processed previously, it is observed that out of 255 pixels, 215 pixels retrieve $\tau_a^{'}$ and 40 pixels do not. Upon analysis of these 40 pixels, it is observed that the these pixels do not converge because the retrieved aerosol optical thicknesses are in excess of 10.0, and DISAMAR stops the retrieval since $\tau$ reaches boundary conditions (beyond 20.0). Such large optical thicknesses may be attributed to the saturation of the top of atmosphere reflectance at very high aerosol loads, observed in Figure 5. It is also possible that these retrievals do not converge because of the presence

of other model errors. Two pixels retrieve $\tau_a^{'}$ above 10.0, and hence are not considered for retrieving $\tau_b^{'}$.

From these 213 pixels, 209 pixels converge to $\tau_b^{'}$, whereas four pixels do not converge to a solution. These four pixels that do not converge are confirmed cases of the presence of aerosol-surface ambiguities, since the retrieval toggles between two values at every iteration until the maximum number of allowable iterations is reached. This is also a consequence of a non-unique top of atmosphere reflectance at high aerosol load scenarios. Out of the 209 pixels that retrieve both $\tau_a^{'}$ and $\tau_b^{'}$, 205 pixels have

similar retrieved optical thickness values according to criterion in Equation 5.2. The rest have values which are off by more than 2.0.

From Figure 8 (right), pixels that contain aerosol-surface ambiguities primarily lie within the main aerosol plume. This is in-line with our expectation of the top of atmosphere being saturated at very high aerosol loads. Interestingly, these pixels also comprise 50% of the pixels that do not converge for aerosol layer height retrieval. Figure 9 (bottom) provides a plot of the

retrieval of CALIPSO co-located GOME-2A pixels, in which 22 pixels are absent from the plot (relative to Figure 9, top). These are pixels for which the prefit method retrieves different $\tau_a^{'}$ and $\tau_b^{'}$.

## 5.3 Discussion

Out of the 100 pixels that do not converge, 50 pixels have been identified which may be affected by aerosol-surface ambiguities. For a majority of these pixels, the retrieved aerosol optical thickness is typically beyond 4.0, for which we can expect



multiple minima in the cost function. It is possible that the true number of pixels that are affected by aerosol-surface ambiguities are higher than 50 pixels — our analysis is represented by a similarity criterion which relies on a similarity threshold $T$, which we have set at 15% (Equation 5.2). With a more strict criterion, more pixels affected by aerosol-surface ambiguities may be detected. Other non-convergences may be a result of model errors. Comparing our retrievals with the CALIOP

attenuated backscatter profile from the infrared channel, we observe that our retrievals are biased closer to the surface, with non-convergences occurring for pixels within the primary biomass burning plume.

## 6   Conclusions

There exists an interference between scattered light by aerosols and the surface to the top of atmosphere reflectance in the oxygen A-band. Our basis for this assertion depends on the distinction of aerosol information present in atmospheric path con-

tributions $R_p$ and surface contributions $R_s$ to the top of atmosphere reflectance in the spectrum (Figure 2). These interferences are dominant in high surface albedo regimes and viewing geometries close to the nadir. A consequence of this interference is a reduction in the amount of information the oxygen A-band spectra has on aerosol parameters.

Our analyses reveal that the information available on aerosol optical thickness in atmospheric path and surface contribution are anti-correlated (see Figure 3, middle), which affects the derivative of reflectance with respect to aerosol optical thickness

changes (see Figure 4). As the surface gets brighter, the magnitude of this derivative decreases, which reduces the sensitivity of the oxygen A-band spectrum to aerosol optical thickness. We expect this anti-correlation behaviour to be strong for viewing angles closer to the nadir, since the interference between $R_p$ and $R_s$ reduces with an increase in viewing angle (see Figure 2). One of the consequences of this interference is the effect on cost function for retrieving aerosol optical thickness. We report that the gradient of the cost function tends to become more shallow as the surface albedo increases. This is especially the case

when the viewing angle is closer to the nadir (see Figure 6). We also notice the presence of multiple minima in the cost function for high aerosol optical thickness values. We attribute this behaviour to the saturation of the top of atmosphere reflectance at high aerosol loads (see Figure 5).

Similar analyses on the available information on aerosol layer height in $R_p$ and $R_s$ in the oxygen A-band reveals that parts of the oxygen A-band spectrum with low absorption by oxygen have interference between $R_p$ and $R_s$ (see Figure 3, left),

which are primarily prevalent over bright surfaces (such as land). It is also observed that the information content available on aerosol single scattering albedo ($\omega$) in $R_p$ and $R_s$ are positively correlated see Figure 3, right), which increases the overall sensitivity of the oxygen A-band spectrum to $\omega$ with increasing surface albedo. This is observed in the derivative of reflectance with respect to $\omega$, which increases in magnitude with an increase in surface albedo.

This interference of $R_p$ and $R_s$ has direct consequences to the retrieval of aerosol layer height from the oxygen A-band.

Over bright surfaces, the retrieval algorithm becomes increasingly susceptible to errors in the aerosol layer height estimates as well as non-convergences in the presence of model errors (see Figure 7). The interference of $R_p$ and $R_s$ also explains why retrieving a aerosol layer over bright surfaces with a 50 hPa thickness for thicker layer (say 200 hPa thickness) can be biased closer to the ground (see Figure 7, top left). To demonstrate this assertion in a real retrieval scenario, we have retrieved aerosol





layer height over the 2010 Russian wildfires in the 8th of August, 2010, using measured oxygen A-band spectra recorded by the GOME-2 instrument on board the Metop-A satellite. For validating our retrievals, we refer to lidar measurements by the CALIOP instrument on board the CALIPSO mission which records, among other measurements, attenuated backscatter at 1064 nm over the same wildfires scene a few hours after the GOME-2A acquisition. Comparison of co-located GOME-2A

5    and CALIPSO pixels reveals that, in the case of both boundary and elevated aerosol layers, the retrieved aerosol layer height is biased closer to the surface. For pixels with a high aerosol load, the algorithm fails to converge to a solution (see Figure 8). Over optically thick plumes, the retrieval becomes dependent on the a-priori aerosol optical thickness (see Figure 8, right). This is, again, a consequence of the interference between $R_p$ and $R_s$.

Following the work presented in this paper, our further goal is to apply the knowledge gained from this study in the devel-
10    opment of the aerosol layer height retrieval algorithm for retrieving aerosols over land.

*Acknowledgements.* This research is partly funded by the European Space Agency (ESA) within the EU Copernicus programme under the project name 'Sentinel-4 Level-2 Processor Component Development', number AO/1-7845/14/NL/MP. We acknowledge EUMETSAT for providing the GOME-2 L1b data.

*Competing interests.* The author declares no conflict of interests in the work expressed in this publication.





## Figures

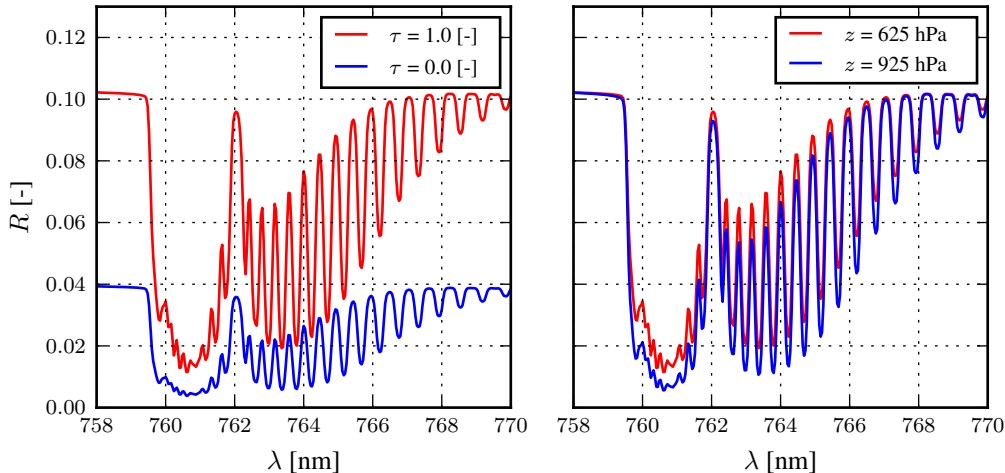

**Figure 1.** Synthetic oxygen A-band spectra for a cloudless atmosphere containing aerosols over a surface with an albedo of 0.03, as measured by a nadir pointing instrument for a solar zenith angle at $45°$. The instrument settings are that of the UVN instrument. Aerosol single scattering albedo is fixed at 0.95 and scattering by aerosols is described by a Henyey-Greenstein phase function with an asymmetry factor ($g$) of 0.7. **Left**: Aerosol layer is fixed at a height of 900 hPa - 950 hPa, for two scenes are different aerosol optical thicknesses. **Right**: Aerosol vertical distribution is varied for an aerosol optical thickness of 1.0 at 760 nm.





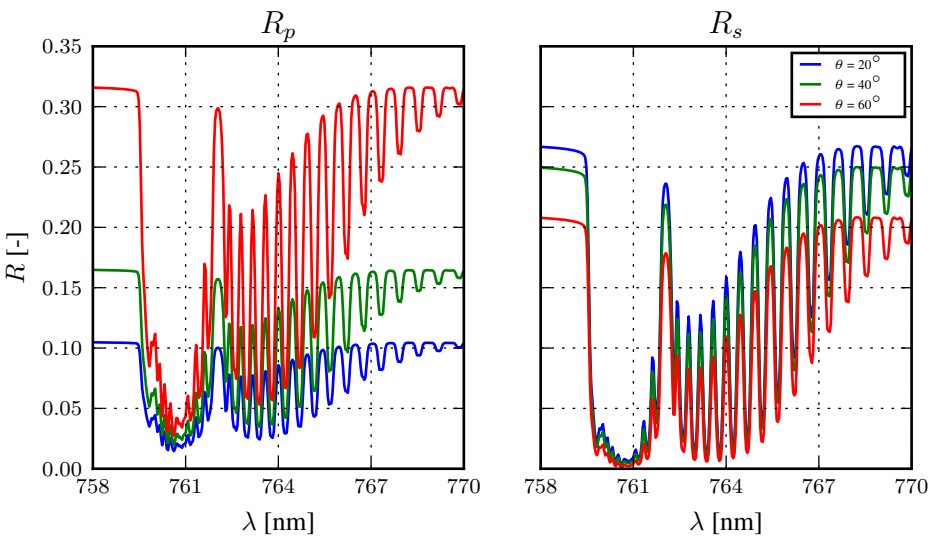

**Figure 2.** $R_p$ and $R_s$ for increasing viewing zenith angle $\theta$ over a surface with an albedo of 0.4 at 760 nm. The solar zenith angle is fixed at 45° and a relative azimuth angle of 0°. Aerosol optical thickness is fixed at 1.0 for an aerosol single scattering albedo of 0.95. Aerosol scattering phase function is a Henyey-Greenstein with $g = 0.7$. The aerosol layer is situated at 600 hPa, with a thickness of 50 hPa.



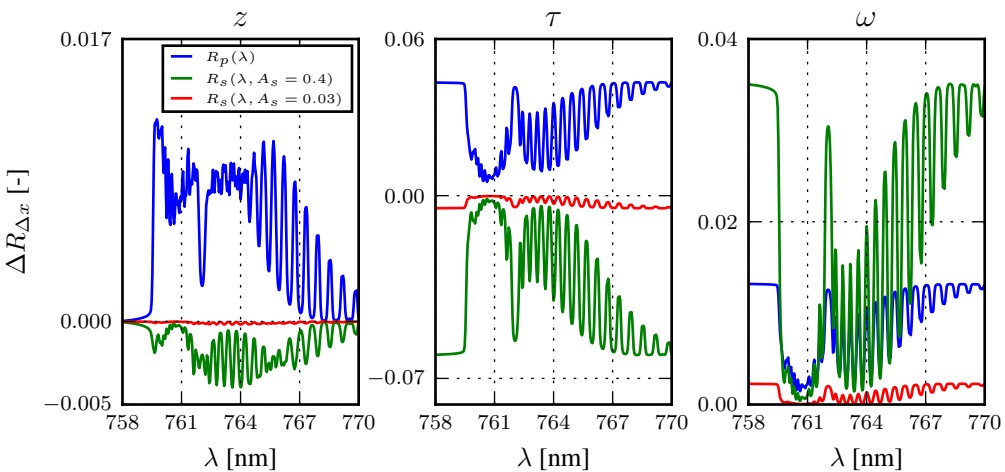

**Figure 3.** $\Delta R_{p_{\Delta x}}$ (in blue) and $\Delta R_{s_{\Delta x}}$ (in red for $A_s = 0.03$ and green for $A_s = 0.4$) to model parameter $x$ in the oxygen A-band, as measured by a nadir pointing instrument for a solar zenith angle at 45°. $\Delta R_{p_{\Delta x}}$ is calculated as the difference of the modeled top-of-atmosphere reflectance between two atmospheres, both cloudless and contain aerosols, which differ only in the parameter $x$ for values $x_a$ and $x_b$, according to Equation 6. The phase function is described by a Henyey-Greenstein model with an anisotropy factor of 0.7, and the thickness of the aerosol layer is fixed at 50 hPa. **Left**: $\tau = 1.0$ and $\omega = 0.95$ with different aerosol layer heights, $z_a = 600$ hPa and $z_b = 800$ hPa. **Middle**: $\tau_a = 1.0$ and $\tau_b = 0.5$ at $z = 600$ hPa and $\omega = 0.95$. **Right**: $\tau = 1.0$ and $z = 600$ hPa for $\omega_a = 0.95$ and $\omega_b = 0.9$. Y-axis has optimised per plot.





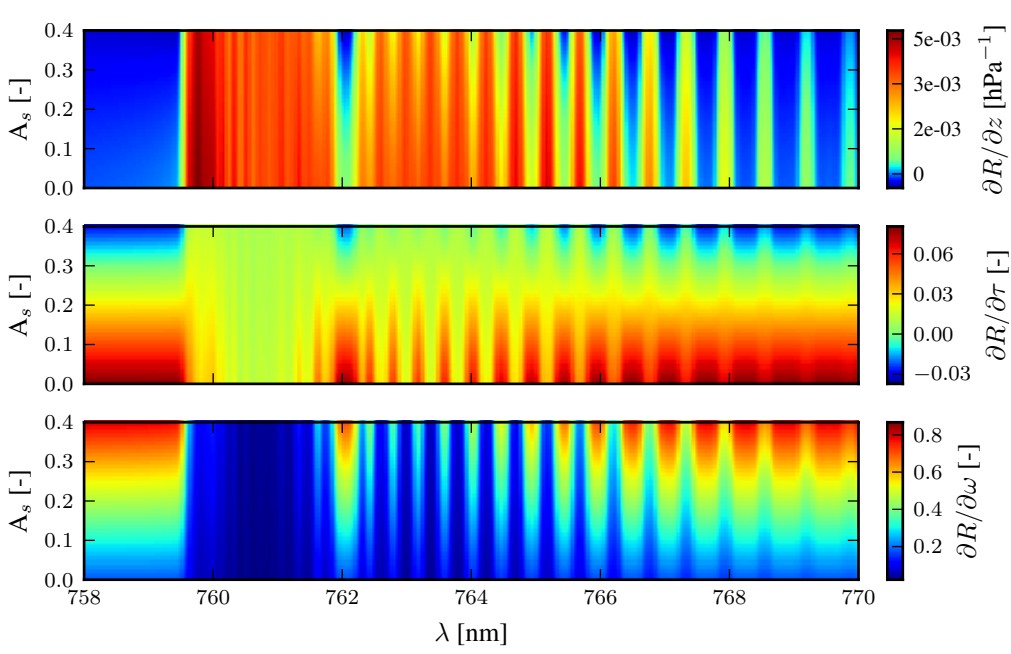

**Figure 4.** Derivative of reflectance with respect to aerosol properties for different surface albedos $A_s$. The $z$ is centered around 600 hPa, with $\tau = 1.0$, $\omega = 0.95$, and a Henyey-Greenstein phase function with $g = 0.7$. The solar zenith angle is $45°$ and the viewing zenith angle is $0°$. **Top**: derivative of reflectance with respect to $z$. **Middle**: derivative of reflectance with respect to $\tau$. **Bottom**: derivative of reflectance with respect to $\omega$. The colorbar has optimised per plot.



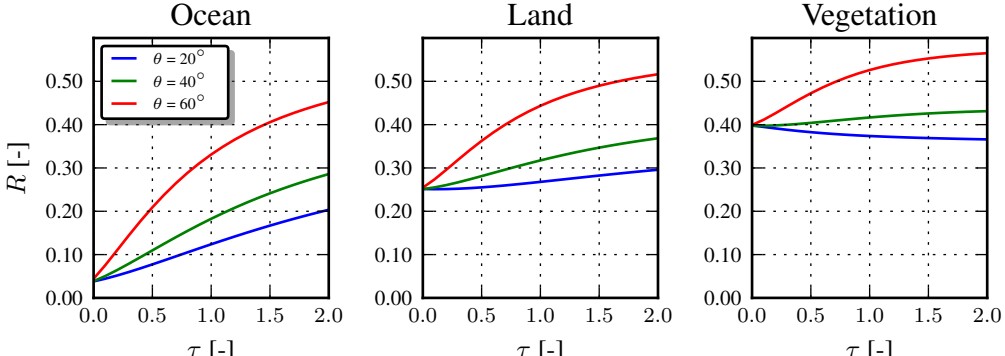

**Figure 5.** Top-of-atmosphere reflectance at 755 nm, well outside the oxygen A-band, from simulated spectra of scenes containing aerosols over dark and bright surfaces. Red, blue and green lines represent different viewing zenith angles $\theta$, as a function of increasing aerosol optical thickness. Aerosols have a single scattering albedo of 0.95, and the aerosol scattering is described by a Henyey-Greenstein phase function with $g = 0.7$. Aerosol layer is situated at 925 hPa. The solar zenith angle is $45°$ and a relative azimuth angle is $0°$. **Left**: The surface albedo is 0.03 at 760 nm, typical over the ocean. **Middle**: The surface albedo is 0.25 at 760 nm, typical over land. **Right**: The surface albedo is 0.4 at 760 nm, typical over vegetated land.

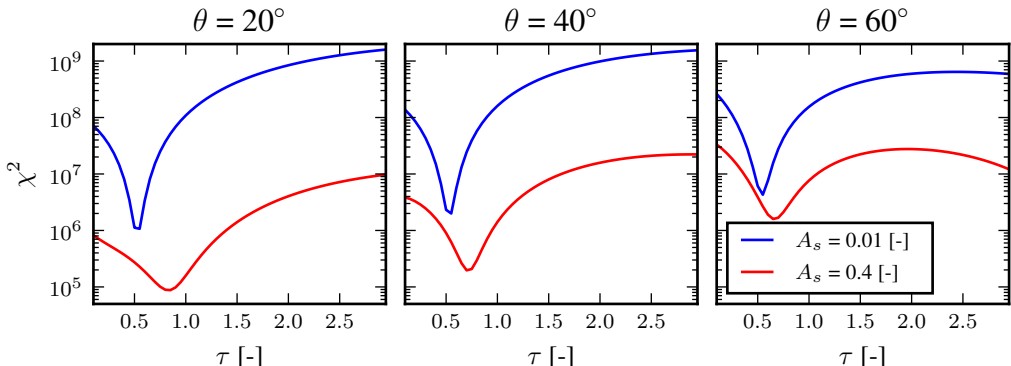

**Figure 6.** Cost function ($\chi^2$) for retrieving aerosol optical thickness as a function of aerosol optical thickness per iteration ($\tau$) for a dark and a bright surface. The true aerosol optical thickness is 0.5, and the aerosol layer is situated at 600 hPa with a 50 hPa layer thickness. The aerosol single scattering albedo is fixed at 0.95, for a Henyey-Greenstein aerosol phase function with $g = 0.7$. The solar zenith angle is fixed at $45°$ for varying viewing angles as specified in the plot titles. The relative azimuth angle is $0°$. The state vector also contains aerosol layer height, whose a-priori value is fixed at 700 hPa.





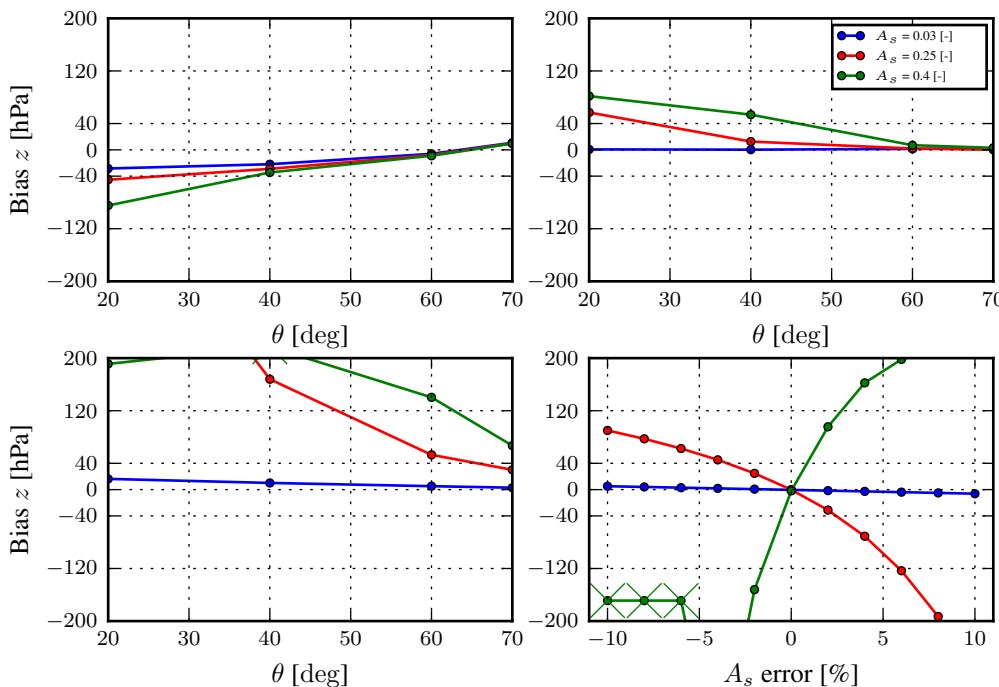

**Figure 7.** Bias in aerosol layer height in the presence of model errors. Unless specified, the relative azimuth angle is $0°$ and the solar zenith angle is $45°$, aerosol single scattering albedo of 0.95 and Henyey-Greenstein $g$ of 0.7, and an aerosol layer at 650 hPa. **Top left**: Model error is introduced in the thickness of the aerosol layer. The simulated spectra contains a 200 hPa thick aerosol plume extending from the 1000 hPa to 800 hPa. **Top right**: Model error is introduced in the aerosol phase function. The simulated scenes contain aerosols with scattering physics described by a Henyey-Greenstein phase function with $g = 0.65$ and retrieved with $g = 0.7$. **Bottom left**: Model error is introduced in the single scattering albedo. The simulated spectra contains aerosols with $\omega = 0.95$, which is fixed in the retrieval forward model at 0.90. **Bottom right**: A relative error is introduced in the surface albedo. The viewing angle is fixed at $20°$.





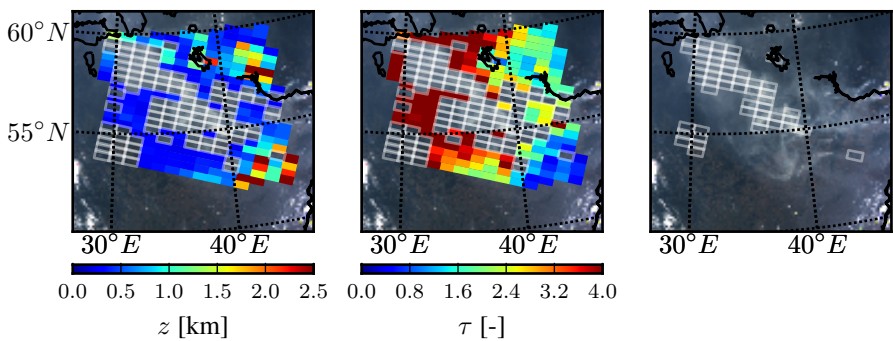

**Figure 8. Left**: Retrieved aerosol layer height from GOME-2A measurements of the 2010 Russian wildfires, in kilometers above the ground with the aerosol layer height retrieval algorithm. Empty white boxes represent pixels that do not converge to a solution. **Middle**: Retrieved aerosol optical thickness from the same retrievals. **Right**: GOME-2A pixels for which there exist possible aerosol-surface ambiguities (empty pixels with white borders).





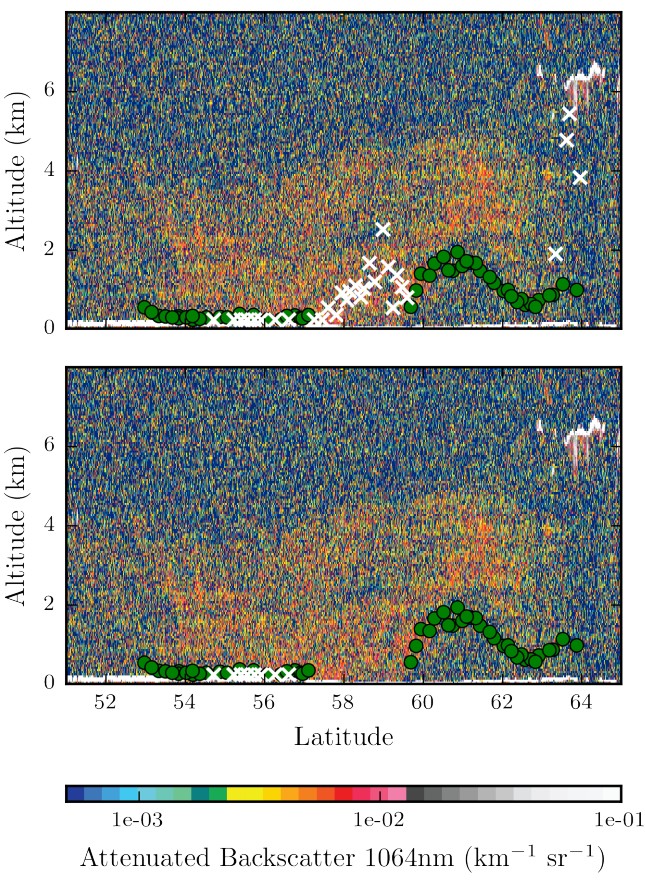

**Figure 9.** CALIOP lidar backscatter cross-section of a track falling within the region of interest over the 2010 Russian wildfire plume on 8[th] of August, 2010. **Top**: green dots and white crosses are GOME-2A pixels falling within 100 km of the CALIPSO ground track — green dots represent converged aerosol layer heights, and white crosses represent the aerosol layer heights at the last iteration for pixels that do not converge to a solution. These retrieved altitudes are reported in km above ground surface. **Bottom**: Retrieval results are presented for pixels for which the the prefit method retrieves both $\tau_a'$ and $\tau_b'$ at similar values.



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
