# Peer review of "Error sources in the retrieval of aerosol information over bright surfaces from satellite measurements in the oxygen A-band"

_Atmospheric Measurement Techniques, 2017_

## Referee Comment (RC1) · Anonymous Referee #1 · 2 Oct 2017

Review comments on manuscript "Error sources in the retrieval of aerosol information over bright surfaces from satellite measurements in the oxygen A-band"

Author(s): Swadhin Nanda et al.

MS No.: amt-2017-323

MS Type: Research article

General comments:

This paper presents a comprehensive analysis on the potential error sources in aerosol height and optical thickness retrieval over different surface brightness. By breaking

down the top of atmosphere reflectance into contribution from surface (Rs) and from the atmospheric column (Rp), the authors are able to analyze the mechanisms behind the potential retrieval bias/errors. The manuscript is organized and presented very well. It provides the community a detailed documentation on the problems facing aerosol property retrieval with oxygen A-band observations. I definitely recommend publication of the paper. I have several minor comments for the authors to consider.

One general suggestion I have is to add some discussions on TOA reflectance sensitibity to layer height for optically thin aerosols. The manuscript uses thick aerosol layers (tau = 1.0) for this purpose (Figure 3 left panel). Since most aerosol layers are optically thin, a similar figure with aerosol optical thickness of 0.2 maybe more telling. I'm thinking a thinner layer would make the problem even harder.

Specific comments:

1) P8, Line 31: Figure 2 does show Rs is more significant than Rp, but that's for albedo = 0.4, not for a dark surface. Maybe just remove "(Figure 2, blue line)".

2) Figure 6: what would be the physical reasons for the retrieval algorithm getting a positively biased optical thickness over brighter surfaces?

3) P11, Line 15: "larger over land than the over the ocean" may be "larger over land that over ocean".

---

## Referee Comment (RC2) · Anonymous Referee #2 · 30 Oct 2017

The manuscript deals with the very challenging retrieval of aerosol height and amount from spectrally resolved satellite observations in the O2A band. An error characterization of a specific retrieval algorithm is presented and its capability is demonstrated in a case study. The topic is very relevant for a number of satellite instruments, the results are new and of interest to the community. Publication in AMT is recommended after the issues listed below are addressed.

Main comments

1. The discussion of the mechanism leading to the near-singular regime (large retrieval uncertainties) is confusing. The authors should avoid talking about 'correlation of in-

formation', 'sign of information', and 'interference between light (contributions Rs and Rp)'. The discussing of the retrieval sensitivity based on the Jacobian (derivative of the top-of-atmosphere radiance R wrt z, AOT, SSA, Fig 4) and the cost function (Fig 6) is clear and instructive. It is not clear what exactly the separation into the derivatives of the contributions Rs and Rp add to this.

2. In many instances the formulations are creative (which is ok as such) but in some cases the clarity suffers. The manuscript needs to be checked and the clarity enhanced.

Minor

1. Page 1 line 22 typo: if an aerosols

2. Page 2 line 10: CALIPSO coverage area is not 'reduced' but maybe 'limited'

3. Page 3 line 6: 'non-consequential' > 'not affected'

4. Page 3 line 13-14: 'the cause of these errors needs to be extended' > 'the concept of critical albedo needs to be extended'

5. Page 5 line 3: formulation: 'due the wavelength band lying beyond' > 'since the wavelength band is located beyond'

6. Page 5 line 10: 'instead of the Henyey-Greenstein MODEL'

7. Page 5 line 18: 'the instrument's platform .. has been designed as a sounder' > 'the instrument is a sounder'

8. Page 5 line 18: remove redundant information in 'The NEAR INFRARED spectrometer . . ., in the NEAR INFRARED.'

9. Page 6 line 6: provide justification for diagonality Page 6 line 16: It is wrong to state that the Jacobian is the primary reason for failure. It is singularity of the generalized inverse.

10. Page 6 line 10: estimate . . . elements .. beyond boundary conditions > beyond boundaries

11. Page 6 line 19: 'reveals interference between Delta_Rp and Delta_Rs' > the increments can cancel.

12. Page 6 line 20: what is the 'relative difference'?

13. Page 8 line 26: the root cause is the cancellation of the increments Delta_Rp and Delta_Rs (same spectral shape, same amplitude, opposite sign) rather than 'anti-correlation'.

14. Page 10 line 14/15: the retrieval sensitivity is not specific to any spectral point > suggested to remove 'at that spectral point' in line 15.

15. Section 4.2. please mention which biases are discussed: bias in AOT or in z, or both?

16. Page 11 line 1 typo: a biases cause by

17. Page 12 line 12/13: Are the terms 'a-priori' and 'first guess' used synonymously? (They should not.) Please report a-priori values for both parameters, or discuss why the first guess is important in this discussion.

18. Page 13 line 4-6: it is stated that the analysis and specifically Fig 7 top left explains the low bias of the retrieved layer height. This is not understood. Please explain.

19. Page 14 line 29 (and Page 15 line 21) it is argued that there are multiple minima in the cost function, in case of high optical thickness. This finding should be presented in the body of the article before it is referred to in the discussion and in the conclusion.

20. Conclusion: please reformulate the discussion of interference and correlation of information.

---

## Author Comment (AC1) · 24 Nov 2017

**Reviewer comment (general):** This paper presents a comprehensive analysis on the potential error sources in aerosol height and optical thickness retrieval over different surface brightness. By breaking down the top of atmosphere reflectance into contribution from surface (Rs) and from the atmospheric column (Rp), the authors are able to analyze the mechanisms behind the potential retrieval bias/errors. The manuscript is organized and presented very well. It provides the community a detailed documentation on the problems facing aerosol property retrieval with oxygen A-band observations. I definitely recommend publication of the paper. I have several minor comments for the

authors to consider.

One general suggestion I have is to add some discussions on TOA reflectance sensitibity to layer height for optically thin aerosols. The manuscript uses thick aerosol layers (tau = 1.0) for this purpose (Figure 3 left panel). Since most aerosol layers are optically thin, a similar figure with aerosol optical thickness of 0.2 maybe more telling. I'm thinking a thinner layer would make the problem even harder.

**Author's response:** Thank you for your constructive feedback. The sensitivity studies in this paper were designed for the KNMI aerosol layer height algorithm, which primarily focuses on aerosol events with a UV absorbing index greater than 2. These are usually quite optically thick layers in the oxygen A-band. However, the question posed is a very valid one - How does the optical thickness determine the accuracy of retrieving aerosol layer height over bright surfaces?

To answer this, we have repeated the sensitivity analyses presented in Figure 7 (bottom left) for retrieval behaviour in the presence of model error in the surface albedo over a bright surface with a surface albedo 0.25 at 760 nm. The simulations are conducted for an aerosol optical thickness of 0.2, 0.5, 0.8 and 1.0. These are more realistic scenarios over Europe in terms of the presence of model errors and the amount of aerosols (see the plot in Figure 1). The caption of the figure is described as follows (the AMT online response editor's figure caption text box does not have enough space for the full caption):

'Bias in retrieved aerosol layer mid pressure (for different aerosol loads) versus introduced model error in the surface albedo relative to the truth (expressed in %) over a scene with a surface albedo of 0.25, viewing at a relative azimuth angle of $0°$, solar zenith angle of $45°$, and viewing zenith angle of $20°$. The aerosols are represented by a single layer present at 650 hPa over a scene with surface pressure of 1013 hPa. The aerosols have a single scattering albedo of 0.95, and the scattering phase function is described by a Henyey-Greenstein model with an anisotropy factor of 0.7. The aerosol

optical thickness is described in the figure legend. Plot points represented with a cross represent non-convergences, i.e., no aerosol layer was retrieved in these cases. The bias is defined as the retrieved aerosol layer height minus the true aerosol layer height — a positive bias suggests that the retrieved aerosol layer is below the true aerosol layer.'

Optically thin aerosol layers will allow more photons to pass through and interact with the surface, leading to an increase in $R_s$, and hence an increase in the cancellation between $R_p$ and $R_s$. With an increase in the aerosol optical thickness, less photons pass through the aerosol layer; consequently, biases in the retrieved aerosol layer height reduce as observed in Figure 1.

**Changes to the manuscript:** We will include the discussion presented here into the manuscript in Section 4.4 as a continuation from line 18 as - 'Retrieving height of optically thin aerosol layers can also be quite challenging, owing to the fact that these layers will allow more photons to pass through and interact with the surface, leading to an increase in $R_s$, and hence an increase in the cancellation between $R_p$ and $R_s$. As a result of this, large biases in the retrieved aerosol layer height can be expected for optically thin layers over bright surfaces.'

**Reviewer comment (specific 1):** P8, Line 31: Figure 2 does show Rs is more significant than Rp, but that's for albedo = 0.4, not for a dark surface. Maybe just remove "(Figure 2, blue line)".

**Author's response:** Agreed.

**Changes to the manuscript:** Removed '(Figure 2, blue line)' from page 8, line 31.

**Reviewer comment (specific 2):** Figure 6: What would be the physical reasons for the

retrieval algorithm getting a positively biased optical thickness over brighter surfaces.

**Author's response:** The bias in the minimum of the cost function, in the case of Figure 6, need not be positive. It is also possible for this bias to be negative. In the calculation of the cost function (Equation 2 in the submitted manuscript), the synthetic measured spectrum **y** (or the 'truth' in this case) is at 600 hPa, whereas the modeled spectrum **F(x,b)** has an aerosol layer at 700 hPa. Because the aerosol layer in the modeled spectrum is lower than the same in the measured spectrum, the modeled atmospheric column that the photon passes through is longer, which increases the amount of absorption by molecular oxygen. Because of this, the aerosol optical thickness in the retrieval forward model is automatically adjusted (in this case, increased) in the minimum in order to compensate for the residual absorption. If the aerosol layer in the model is higher in the atmosphere than the same in the 'measured' (synthetic truth) spectrum, the minimum will appropriately shift to a lower value, in order to compensate for a deficit in the absorption in the spectrum generated by the retrieval forward model.

**Changes to the manuscript:** We have changed Page 9 line 9 (second paragraph) from 'Also, as Rs increases, the global minimum shifts away from the true $\tau$. This is predominantly observed in Figure 6 (left, red line) over the bright surface for a viewing angle close to nadir, where $R_s$ is more dominant. For the same angle, the global minimum over a dark surface is situated at the true $\tau$ value. As the viewing angle increases over the bright surface, $R_p$ increases and the global minimum of the cost function moves closer towards the true $\tau$.' to the following:

'Also, because of a model error (described in Figure 6) in the aerosol layer height between **y** and **F(x,b)**, (in Equation 2) the global minimum of the cost functions shifts away from the true $\tau$. This shift is baised higher than the truth if the aerosol layer is lower in the atmosphere in comparison to the aerosol layer in the synthetic true spectrum, because the model has to compensate the extra absorption by molecular oxygen. If the aerosol layer is higher in the atmosphere, the minimum of the cost

function is situated at a $\tau$ lower than the true $\tau$. As observed in Figure 6 (left, red line), this shift of the cost function minimum from the true $\tau$ is larger over bright surfaces for a viewing angle close to nadir, where $R_S$ is more dominant. For the same angle, the global minimum over a dark surface is situated at the true $\tau$ value, even with the presence of a model disagreement with the simulated 'true' spectrum. As the viewing angle increases over the bright surface, $R_p$ increases and the global minimum of the cost function moves closer towards the true $\tau$.'

**Reviewer comment (specific 3):** P11, Line 15: "larger over land than the over the ocean" may be "larger over land that over ocean".

**Author's response:** Agreed.

**Changes to the manuscript:** Page 11, Line 15: changed 'larger over land than the over the ocean" changed to 'larger over land than over ocean".
* * *
[Figure]

Figure showing Bias $p_{mid}$ [hPa] versus $A_s$ error [%] with curves for $\tau = 0.2$ [-], $\tau = 0.5$ [-], $\tau = 0.8$ [-], $\tau = 1.0$ [-].

**Fig. 1.** Please check text for the caption.

---

## Author Comment (AC2) · 24 Nov 2017

**Reviewer comment (general):** The manuscript deals with the very challenging retrieval of aerosol height and amount from spectrally resolved satellite observations in the O2A band. An error characterization of a specific retrieval algorithm is presented and its capability is demonstrated in a case study. The topic is very relevant for a number of satellite instruments, the results are new and of interest to the community. Publication in AMT is recommended after the issues listed below are addressed.

The discussion of the mechanism leading to the near-singular regime (large retrieval uncertainties) is confusing. The authors should avoid talking about 'correlation of information', 'sign of information', and 'interference between light (contributions Rs and Rp)'. The discussing of the retrieval sensitivity based on the Jacobian (derivative of the top-of-atmosphere radiance R wrt z, AOT, SSA, Fig 4) and the cost function (Fig 6) is clear and instructive. It is not clear what exactly the separation into the derivatives of the contributions Rs and Rp add to this.

**Author's response:** Thank you for the constructive comments. Agreed that such terms can become confusing. These terms are appropriately changed and further discussed in the response to the second general comment. We will respond to your comment on our chosen method of explaining the retrieval problem using $R_p$ and $R_s$.

The derivative of the reflectance with respect to model parameters is an important tool in understanding retrieval sensitivities. However, a general understanding of the change in the magnitude of the derivatives in Figure 4 with increasing surface albedo is, in our opinion, better explained as the interaction between $R_p$ and $R_s$. This mathematical splitting of light into atmospheric path and surface contributions explains the reduction in the derivative of the reflectance to aerosol layer height with an increase in surface albedo at specific parts of the oxygen A-band spectrum (around 762 nm in Figure 4 top for instance) as a consequence of increasing $R_s$ at the same spectral points. This also explains why the spectral shape of the derivative of the reflectance with respect of surface albedo becomes relatively more flat at approximately 0.25 surface albedo, with little variation between the continuum at 758 nm and the deepest part of the oxygen A-band in the R branch at 761 nm - the derivatives alone cannot explain why they change. We have, hence, chosen the splitting of top of atmosphere reflectance into individual atmospheric path and surface contributions as a tool to diagnose the variability of the Jacobian with change in model parameters.

**Changes to the manuscript:** Regarding the clarification of the manuscript with more appropriate terminology, we have made changes in-line with the response to the following reviewer general comment. We do not propose any changes regarding the second part of our response.

**Reviewer comment (general):** In many instances the formulations are creative (which is ok as such) but in some cases the clarity suffers. The manuscript needs to be checked and the clarity enhanced.

**Author's response:** We expect that with the proposed changes to the manuscript, the paper is more clear. As proposed by the referee, we will change some 'creative' terms in the manuscript with more apt describers. These are detailed in the following.

**Changes to the manuscript:** Page 1 Abstract, line 9 - changed 'The analysis shows that the information on aerosol layer height from atmospheric path contribution and the surface contribution to the top of atmosphere are opposite in sign' to 'The analysis shows that the derivative, with respect to aerosol layer height, of the atmospheric path contribution to the top-of-atmosphere reflectance is opposite in sign to the same of surface contribution - ...'

Page 7 line 14 - changed 'the difference spectra ...' to 'the difference spectrum'

Page 7 line 19 - changed 'it reveals an interference between these two contributions to the top of atmosphere reflectance' to '$\Delta R_{\Delta x}$ reduces following Equation 7, which may ...'

Page 7 line 28 - changed 'and hence an increase in interference between ...' to 'and hence an increase in cancellation between $\Delta R_{p\Delta z}$ and $\Delta R_{s\Delta z}$.'

Page 8 line 1-2 - changed 'Albeit subtle, the consequence of interference between ...' to 'Albeit subtle, the consequence of this cancellation between $\Delta R_{p\Delta z}$ and $\Delta R_{s\Delta z}$'

Page 8 line 7 - changed 'The interference between $\Delta R_{p\Delta\tau}$ and $\Delta R_{s\Delta\tau}$' to 'This anti-correlation'

Page 10 line 2-3 - changed 'This is explained by the lowered interference between $R_p$ and $R_s$ (Figure 2, red line). This is why the difference in errors between retrievals over the different surfaces reduces with an increase in viewing zenith angle' to 'This is explained by a reduction in $R_s$ and an increase in $R_p$, (Figure 2, red line), which

explains why difference in errors between retrievals over different surfaces reduces with an increase in viewing angle (Figure 7, top left, high viewing zenith angles)'.

Page 10 line 10-26 - changed 'While it would appear that somehow the sensitivity of the retrieval to aerosol layer thickness increases with increasing surface albedo, this behaviour is explained better as perhaps the most interesting consequence of the interference between $R_s$ and $R_p$ ingrained within the optimal estimation framework. ... appear to have more absorption for observations over high surface albedos, corresponding to an aerosol layer closer to the surface than the true aerosol layer height' to 'This should not suggest a sensitivity to the geometrical thickness of the aerosol layer. As the surface albedo increases, the number of photons that pass through the atmosphere to interact with the surface before reaching the detector increases. These photons have a longer path length, which results in an increased absorption by oxygen at specific spectral points with weak oxygen absorption lines. In comparison to photons at wavelengths with strong oxygen absorption lines, these photons have a higher SNR, since relatively more of them reach the detector. A higher SNR ensures lower noise, and hence a higher value in the measurement error covariance matrix $S_\epsilon$. If a spectral point has a higher value in $S_\epsilon$, it has a higher representation in the cost function (in Equation 2), and hence a higher preference (or weight) in the optimal estimation. Because of this, the retrieval prefers to retrieve an aerosol layer height described by photons that travel through the aerosol layer closer to the surface. If, however, the aerosol optical thickness is so large that the photons cannot penetrate the aerosol layer, the retrieved aerosol layer height would be more accurate.'

Page 10 line 27 - Changed 'in the presence of interference between $R_p$ and $R_s$' to 'over bright surfaces'

Page 11 line 8-9 - changed 'This again attributed to the decreased interference between $R_p$ and $R_s$ with increasing viewing angle' to 'This is again attributed to the decrease in $R_p$ and increase in $R_s$ with increasing viewing angle'.

[Figure]

Page 13 line 5 - please check response to minor comment 18

Page 15 line 8 - 'There exists an interference between scattered light by aerosols and the surface to the top of atmosphere reflectance in the oxygen A-band' changed to 'Depending on the surface brightness, the interaction of photons scattered from the atmosphere and the surface can result in a possible reduction of available aerosol information in the oxygen A-band spectrum.'

Page 15 line 10-11 - changed 'These interferences are dominant in high surface albedo regimes and viewing geometries close to the nadir' to 'The reduction of aerosol information increases with increasing surface brightness and decreasing viewing angle'

Page 15 line 11-12 - removed 'A consequence of this interference is a reduction in the amount of information the oxygen A-band spectra has on aerosol parameters.'

Page 15 line 12 - changed 'Our analyses reveal that the information available on aerosol optical ...' to 'Our analyses reveal that the derivatives of the atmospheric path and surface contributions with respect to ...'

Page 15 line 18 - 'since the interference between $R_p$ and $R_s$ reduces' changed to 'since $R_p$ increases and $R_s$ decreases'

Page 15 line 24-25 - 'with low absorption by oxygen have interference between $R_p$ and $R_s$ (see Figure 3, left), which are primarily prevalent over bright surfaces (such as land).' changed to 'with a low absorption by oxygen have an increased cancellation of $\Delta R_{p\Delta z}$ and $\Delta R_{s\Delta z}$ (see Figure 3, left) and hence a reduction in aerosol layer height sensitivity in specific parts of the spectrum (see Figure 4, top). This increases as surface albedo increases.'

Page 15 line 25-26 - 'It is also observed that the information content available on aerosol single scattering albedo ($\omega$) in $R_p$ and $R_s$ are positively correlated' changed to 'it is also observed that the derivative of $\Delta R_{p\Delta\omega}$ and $\Delta R_{s\Delta\omega}$ are both positive (see Figure 3, right)'

Page 15 line 29 - changed 'This interference of $R_p$ and $R_s$' to 'The interaction between photons scattering back from the atmosphere ($R_p$) to the detector and photons that travel through the atmosphere to the surface to the detector ($R_s$)'

Page 15 line 31 - changed 'The interference of $R_p$ and $R_s$' to 'The sign difference of $\Delta R_{p\Delta\tau}$ and $\Delta R_{s\Delta\tau}$'

Page 16 line 8 - Removed 'This is, again, a consequence of the interference between $R_p$ and $R_s$'

**Reviewer comment (minor 1):** Page 1 line 22 typo: if an aerosols

**Author's response:** Accepted.

**Changes to the manuscript:** Changed Page 1 line 22 'if an aerosols' to 'if aerosols'

**Reviewer comment (minor 2):** 2. Page 2 line 10: CALIPSO coverage area is not 'reduced' but maybe 'limited'

**Author's response:** Accepted.

**Changes to the manuscript:** Changed Page 2 line 9-10 from 'However, because of the limited swath of a space-borne lidar instrument, the mission coverage area is significantly reduced' to 'However, because of the limited swath of a space-borne lidar instrument, the mission coverage area is also limited'.

**Reviewer comment (minor 3):** Page 3 line 6: 'non-consequential' > 'not affected'

**Author's response:** Accepted.

**Changes to the manuscript:** Page 3 line 6 'non-consequential' changed to 'not affected'.

**Reviewer comment (minor 4):** Page 3 line 13-14: 'the cause of these errors needs to be extended' > 'the concept of critical albedo needs to be extended'

**Author's response:** Accepted.

**Changes to the manuscript:** Page 3 line 13-14 'the cause of these errors needs to be extended' changed to 'the concept of critical albedo needs to be extended'.

**Reviewer comment (minor 5):** Page 5 line 3: formulation: 'due the wavelength band lying beyond' > 'since the wavelength band is located beyond'

**Author's response:** Accepted.

**Changes to the manuscript:** Page 5 line 3 'due the wavelength band lying beyond' changed to 'since the wavelength band is located beyond'.

**Reviewer comment (minor 6):** Page 5 line 10: 'instead of the Henyey-Greenstein MODEL'

**Author's response:** Accepted.

**Changes to the manuscript:** Page 5 line 10 changed from 'instead of the Henyey-Greenstein' to 'instead of the Henyey-Greenstein model'.

**Reviewer comment (minor 7):** Page 5 line 18: 'the instrument's platform .. has been designed as a sounder' > 'the instrument is a sounder'

**Author's response:** Accepted.

**Changes to the manuscript:** Changed page 5 line 18-19 'The instrument's platform has been designed as a geostationary atmospheric sounder with a hourly coverage' to 'The instrument is an atmospheric sounder on a geostationary platform with an hourly coverage'.

**Reviewer comment (minor 8):** Page 5 line 18: remove redundant information in 'The NEAR INFRARED spectrometer . . ., in the NEAR INFRARED.'

**Author's response:** Accepted with changes.

**Changes to the manuscript:** Page 5 line 20 changed 'The near infrared spectrometer has a FWHM of approximately 0.116 nm in the near infrared, oversampled by a factor of 3' to 'The instrument has a FWHM of approximately 0.116 nm, oversampled by a factor 3'.

**Reviewer comment (minor 9):** Page 6 line 6: provide justification for diagonality Page 6 line 16: It is wrong to state that the Jacobian is the primary reason for failure. It is singularity of the generalized inverse.

**Author's response:** Accepted.

**Changes to the manuscript:** Changed 'The matrices $S_a$ and $S_\epsilon$ are diagonal, and are not correlated' to '$S_a$ is diagonal, assuming no correlation between state vector elements. $S_\epsilon$ is also diagonal, since the measurement error is assumed uncorrelated.'.

Page 6 line 15-17 'The Jacobian is also the primary reason why the retrieval can fail — the Jacobian can become singular if the value of the partial derivative of the reflectance to the a state vector parameter is very low, or is correlated to another parameter in the state vector' changed to 'the Jacobian can become singular if the value of the partial

derivative of the reflectance to the a state vector parameter is very low, or is correlated to another parameter in the state vector.'

**Reviewer comment (minor 10):** Page 6 line 10: estimate . . . elements .. beyond boundary conditions $>$ beyond boundaries

**Author's response:** Accepted.

**Changes to the manuscript:** Page 6 line 21: 'the inverse method estimates state vector elements beyond their physical boundary conditions' changed to 'the inverse method estimates state vector elements beyond boundaries'

**Reviewer comment (minor 11):** Page 6 line 19: 'reveals interference between $\Delta R_p$ and $\Delta R_s$' $>$ the increments can cancel.

**Author's response:** Accepted.

**Changes to the manuscript:** Please check response to main comment 2.

**Reviewer comment (minor 12):** Page 6 line 20: what is the 'relative difference'?

**Author's response:** In context to page 7 line 20, the line 'In such a case, the relative difference of $\Delta R_p$ and $\Delta R_s$ gives an idea on the magnitude of interference.' does not add to the paper very well. We will remove this line, in order to reduce any confusion/redundancy.

**Changes to the manuscript:** Page 7 line 20 removed 'In such a case, the relative difference of $\Delta R_p$ and $\Delta R_s$ gives an idea on the magnitude of interference'.

**Reviewer comment (minor 13):** Page 8 line 26: the root cause is the cancellation of the increments Delta_Rp and Delta_Rs (same spectral shape, same amplitude, opposite sign) rather than 'anticorrelation'.

**Author's response:** Accepted.

**Changes to the manuscript:** Changed Page 8 line 26 from 'due to the anti-correlation between $\Delta R_p$ and $\Delta R_s$' to 'due to the cancellation between $\Delta R_p$ and $\Delta R_s$ owing to their similar amplitudes, spectral shapes but opposing signs.'

**Reviewer comment (minor 14):** Page 10 line 14/15: the retrieval sensitivity is not specific to any spectral point $>$ suggested to remove 'at that spectral point' in line 15.

**Author's response:** Accepted.

**Changes to the manuscript:** Removed 'at that spectral point' in page 10 line 15.

**Reviewer comment (minor 15):** Section 4.2. please mention which biases are discussed: bias in AOT or in z, or both?

**Author's response:** Section 4 discusses biases in the retrieved aerosol layer height. An extra line in the introduction to section 4 is added to clarify all subsequent mentions of 'biases'.

**Changes to the manuscript:** Section 4 Page 9 line 23-24 changed 'Error analysis is done for the aerosol layer height retrieval algorithm and a comparison is made between retrievals over ocean ...' to 'A comparative analysis of biases in the retrieved aerosol layer height is conducted over ocean ...'

**Reviewer comment (minor 16):** Page 11 line 1 typo: a biases cause by

**Author's response:** Accepted.

**Changes to the manuscript:** Changed Page 11 line 1 'The correlation of bias with surface albedo suggests that a biases cause by model ...' to 'The correlation of bias with surface albedo suggests that biases caused by model ...'

**Reviewer comment (minor 17):** Page 12 line 12/13: Are the terms 'a-priori' and 'first guess' used synonymously? (They should not.) Please report a-priori values for both parameters, or discuss why the first guess is important in this discussion.

**Author's response:** Accepted. Indeed this is a mistake on our part. The terms 'a-priori' and 'first guess' are not the same.

Also, the LER database used in the retrievals conducted in this paper are not on a 0.5° x 0.5° grid. Rather they are on a 1° x 1° grid.

**Changes to the manuscript:** Changed Page 12 line 10 'Surface albedo is derived using nearest neighbour interpolation from Tilstra et al. (2017), who provide monthly Lambertian Equivalent Reflectivity (LER) climatologies on a 0.5° x 0.5° grid' to 'Surface albedo is derived using nearest neighbour interpolation for version 1.3 of GOME-2A LER climatology derived from Tilstra et al. (2017), which is at a 1° x 1° grid'.

Changed Page 12 line 13 'In the inverse method, the first guess ...' to 'In the inverse method, the a-priori value ...'

Changed Page 13, Table 1 entry for 'surface albedo As' from 'GOME-2 LER at 0.5° x 0.5° grid at 758 nm and 772 nm' to 'GOME-2A LER at 1° x 1° grid at 758 nm and 772 nm'.

**Reviewer comment (minor 18):** Page 13 line 4-6: it is stated that the analysis and specifically Fig 7 top left explains the low bias of the retrieved layer height. This is not

understood. Please explain.

**Author's response:** Figure 7 (top left) provides examples of retrieving an aerosol layer of 50 hPa geometric thickness for a spectrum that represents an aerosol layer with a 200 hPa geometric thickness at an aerosol optical thickness of 1.0 at 760 nm. We observe that over higher surface albedos, the retrieved aerosol layer mid height is biased closer to the surface. Over a higher surface albedo, the contribution by $R_s$ is also greater. These biases reduce significantly as the viewing zenith angle increases, which is also when $R_s$ reduces (according to Figure 2). Since this is discussed in more detail in 4.1, we will mention a brief version of it in the changed version.

We observed, post submission, that the definition of bias in aerosol layer height mentioned in Page 9, line 25-26 is incorrect. This only applies to Figure 7 top left, and not to the rest of the sensitivity studies. For this we will include the correct figure (Figure 1) with the following caption

'Bias in aerosol layer height in the presence of model errors. Unless specified, the relative azimuth angle is $0°$ and the solar zenith angle is $45°$, aerosol single scattering albedo of 0.95 and Henyey-Greenstein $g$ of 0.7, and an aerosol layer at 650 hPa. **Top left**: Model error is introduced in the thickness of the aerosol layer. The simulated spectra contains a 200 hPa thick aerosol plume extending from the 1000 hPa to 800 hPa. **Top right**: Model error is introduced in the aerosol phase function. The simulated scenes contain aerosols with scattering physics described by a Henyey-Greenstein phase function with $g = 0.65$ and retrieved with $g = 0.7$. **Bottom left**: Model error is introduced in the single scattering albedo. The simulated spectra contains aerosols with $\omega = 0.95$, which is fixed in the retrieval forward model at 0.90. **Bottom right**: A relative error is introduced in the surface albedo. The viewing angle is fixed at $20°$.'

From these analyses, it is not straightforward to assume that the retrieved aerosol layer will be retrieved closer to the ground if surface albedo in the retrieval model is greater than the true surface albedo — this is not observed in Figure 6 bottom right for a surface

albedo of 0.25 and 0.4.

**Changes to the manuscript:** Changed 'From our analysis in Figure 7 (top left)' to 'This is explained by the increase in surface contribution $R_s$ which represents photons passing through the atmosphere and interacting with the surface before reaching the detector. The spectral points representing these photons have a higher weight in the optimal estimation in comparison to the photons that do not interact with the surface and hence the aerosol layer height is retrieved closer to the surface.'

We propose to replace Figure 7 in the manuscript with Figure 1 proposed in this author's response. We also propose to replace Page 9 line 25-26 'Bias in the aerosol layer height is defined as the difference between true and retrieved aerosol layer height (in hPa) — a negative sign indicates that the aerosol layer is retrieved closer to the ground' with 'Bias in the aerosol layer height is defined as the difference between retrieved and true aerosol layer height (in hPa) — a positive sign indicates that the aerosol layer is retrieved closer to the ground with respect to the true aerosol layer height.'

**Reviewer comment (minor 19):** Page 14 line 29 (and Page 15 line 21) it is argued that there are multiple minima in the cost function, in case of high optical thickness. This finding should be presented in the body of the article before it is referred to in the discussion and in the conclusion.

**Author's response:** Section 3.2 Page 9, line 4 - 12 discusses the implication of surface brightness on the cost function. The general conclusion from Figure 6 is that the cost function can also contain multiple minima, one near the true aerosol optical thickness and one at a much higher value (Figure 6, left, slope after tau = 1.5). However, the figure presents only a single synthetic case of multiple minima over specific solar-satellite geometries. We understand that this can cause confusion in interpreting results presented in the paper. For that matter, we propose to remove the mention of multiple minima from Page 14 line 29. However, in Page 15, line 21, we refer to figure

6 (right), although not explicitly, where we see the cost function decreasing beyond the local minima close to the true aerosol optical thickness. To make this clear, we propose to mention Figure 6 (right) in this line, with the sentence made specific for the case presented in this figure.

**Changes to the manuscript:** Page 14 line 29 changed 'the retrieved aerosol optical thickness is typically beyond 4.0, for which we can expect multiple minima in the cost function' to 'the retrieved aerosol optical thickness is typically beyond 4.0.'

Page 15 line 20-21 changed 'We also notice the presence of multiple minima in the cost function for high aerosol optical thickness values' to 'We also notice that the cost function reduces at high aerosol optical thickness beyond the local minimum near the truth (Figure 6, right), which indicates the presence of multiple minima in the cost function.'

**Reviewer comment (minor 20):** Conclusion: please reformulate the discussion of interference and correlation of information.

**Author's response:** Accepted.

**Changes to the manuscript:** The following changes are also described in response to general comment 2.

Page 15 line 8 - 'There exists an interference between scattered light by aerosols and the surface to the top of atmosphere reflectance in the oxygen A-band' changed to 'Depending on the surface brightness, the interaction of photons scattered from the atmosphere and the surface can result in a possible reduction of available aerosol information in the oxygen A-band spectrum.'

Page 15 line 10-11 - changed 'These interferences are dominant in high surface albedo regimes and viewing geometries close to the nadir' to 'The reduction of aerosol information increases with increasing surface brightness and decreasing viewing angle'

Page 15 line 11-12 - removed 'A consequence of this interference is a reduction in the amount of information the oxygen A-band spectra has on aerosol parameters.'

Page 15 line 12 - changed 'Our analyses reveal that the information available on aerosol optical ...' to 'Our analyses reveal that the derivatives of the atmospheric path and surface contributions with respect to ...'

Page 15 line 18 - 'since the interference between $R_p$ and $R_s$ reduces' changed to 'since $R_p$ increases and $R_s$ decreases'

Page 15 line 24-25 - 'with low absorption by oxygen have interference between $R_p$ and $R_s$ (see Figure 3, left), which are primarily prevalent over bright surfaces (such as land).' changed to 'with a low absorption by oxygen have an increased cancellation of $\Delta R_{p\Delta z}$ and $\Delta R_{s\Delta z}$ (see Figure 3, left) and hence a reduction in aerosol layer height sensitivity in specific parts of the spectrum (see Figure 4, top). This increases as surface albedo increases.'

Page 15 line 25-26 - 'It is also observed that the information content available on aerosol single scattering albedo ($\omega$) in $R_p$ and $R_s$ are positively correlated' changed to 'it is also observed that the derivative of $\Delta R_{p\Delta\omega}$ and $\Delta R_{s\Delta\omega}$ are both positive (see Figure 3, right)'

Page 15 line 29 - changed 'This interference of $R_p$ and $R_s$ ' to 'The interaction between photons scattering back from the atmosphere ($R_p$) to the detector and photons that travel through the atmosphere to the surface to the detector ($R_s$)'

Page 15 line 31 - changed 'The interference of $R_p$ and $R_s$ ' to 'The sign difference of $\Delta R_{p\Delta\tau}$ and $\Delta R_{s\Delta\tau}$'

Page 16 line 8 - Removed 'This is, again, a consequence of the interference between $R_p$ and $R_s$ '

[Figure]

Fig. 1. Caption in text (response to minor 18)